# Distinct higher-order representations of natural sounds in human and ferret auditory cortex

**Agnès Landemard[1]\*[†], Célian Bimbard[1,2][†], Charlie Demené[3], Shihab Shamma[1,4], Sam Norman-Haignere[1,5,6][‡], Yves Boubenec[1]\*[‡]**

[1]Laboratoire des Systèmes Perceptifs, Département d'Études Cognitives, École Normale Supérieure PSL Research University, CNRS, Paris, France; [2]University College London, London, United Kingdom; [3]Physics for Medicine Paris, Inserm, ESPCI Paris, PSL Research University, CNRS, Paris, France; [4]Institute for Systems Research, Department of Electrical and Computer Engineering, University of Maryland, College Park, United States; [5]HHMI Postdoctoral Fellow of the Life Sciences Research Foundation, Baltimore, United States; [6]Zuckerman Mind Brain Behavior Institute, Columbia University, New York, United States

**\*For correspondence:**
agnes.landemard@ens.fr (AL);
yves.boubenec@ens.fr (YB)

[†]These authors contributed equally to this work
[‡]These authors also contributed equally to this work

**Competing interest:** The authors declare that no competing interests exist.

**Abstract** Little is known about how neural representations of natural sounds differ across species. For example, speech and music play a unique role in human hearing, yet it is unclear how auditory representations of speech and music differ between humans and other animals. Using functional ultrasound imaging, we measured responses in ferrets to a set of natural and spectrotemporally matched synthetic sounds previously tested in humans. Ferrets showed similar lower-level frequency and modulation tuning to that observed in humans. But while humans showed substantially larger responses to natural vs. synthetic speech and music in non-primary regions, ferret responses to natural and synthetic sounds were closely matched throughout primary and non-primary auditory cortex, even when tested with ferret vocalizations. This finding reveals that auditory representations in humans and ferrets diverge sharply at late stages of cortical processing, potentially driven by higher-order processing demands in speech and music.

## Editor's evaluation

How the auditory system encodes speech sounds is not well understood, and animal models have a lot to offer in investigating such questions. This study evaluated the representations of a variety of natural and synthetic sounds in both ferrets and humans, and reported that humans differed from ferrets in the manner in which speech and music were represented, despite controlling for the spectrotemporal content of the sounds. This work makes an important contribution to our understanding of how the coding of such sounds differs across species.

## Introduction

Surprisingly little is known about how sensory representations of natural stimuli differ across species (*Theunissen and Elie, 2014*). This question is central to understanding how evolution and development shape sensory representations (*Moore and Woolley, 2019*) as well as developing animal models of human brain functions. Audition provides a natural test case because speech and music play a unique role in human hearing (*Zatorre et al., 2002*; *Hickok and Poeppel, 2007*; *Patel, 2012*). While human knowledge of speech and music clearly differs from other species (*Pinker and Jackendoff,*

*2005*), it remains unclear how neural representations of speech and music differ from those in other species, particularly within the auditory cortex. Few studies have directly compared neural responses to natural sounds between humans and other animals, and those that have done so have often observed similar responses. For example, both humans and non-human primates show regions that respond preferentially to conspecific vocalizations (*Belin et al., 2000*; *Petkov et al., 2008*). Human auditory cortex exhibits preferential responses for speech phonemes (*Mesgarani et al., 2014*; *Di Liberto et al., 2015*), but much of this sensitivity can be predicted by simple forms of spectrotemporal modulation tuning (*Mesgarani et al., 2014*), and perhaps as a consequence can be observed in other animals such as ferrets (*Mesgarani et al., 2008*; *Steinschneider et al., 2013*). Consistent with this finding, maps of spectrotemporal modulation, measured using natural sounds, appear coarsely similar between humans and macaques (*Erb et al., 2019*), although temporal modulations present in speech may be over-represented in humans. Thus, it remains unclear if the representation of natural sounds in auditory cortex differs substantially between humans and other animals, and if so, how.

A key challenge is that representations of natural stimuli are transformed across different stages of sensory processing, and species may share some but not all representational stages. Moreover, responses at different sensory stages are often correlated across natural stimuli (*de Heer et al., 2017*), making them difficult to disentangle. Speech and music, for example, have distinctive patterns of spectrotemporal modulation energy (*Singh and Theunissen, 2003*; *Ding et al., 2017*), as well as higher-order structure (e.g., syllabic and harmonic structure) that is not well captured by modulation (*Norman-Haignere et al., 2018*). To isolate neural sensitivity for higher-order structure, we recently developed a method for synthesizing sounds whose spectrotemporal modulation statistics are closely matched to a corresponding set of natural sounds (*Norman-Haignere et al., 2018*). Because the synthetic sounds are otherwise unconstrained, they lack perceptually salient higher-order structure, which is particularly true for complex natural sounds like speech and music that are poorly captured by modulation statistics, unlike many other natural sounds (*McDermott and Simoncelli, 2011*). We found that human primary auditory cortex responds similarly to natural and spectrotemporally matched synthetic sounds, while non-primary regions respond preferentially to the natural sounds. Most of this response enhancement is driven by preferential responses to natural vs. synthetic speech and music in non-primary auditory cortex. The specificity for speech and music could be due to their ecological relevance in humans and/or the fact that speech and music are more complex than other sounds, and thus perceptually differ more from their synthetic counterparts. But notably, the response preference for natural speech and music cannot be explained by speech semantics since similar responses are observed for native and foreign speech (*Norman-Haignere et al., 2015*; *Overath et al., 2015*), or explicit musical training, since humans without any training show similar response preferences for music in their non-primary auditory cortex (*Boebinger et al., 2020*). These findings suggest that human non-primary regions respond to higher-order acoustic features that both cannot be explained by lower-level modulation statistics and do not yet reflect explicit semantic knowledge.

The goal of the present study was to test whether such higher-order sensitivity is present in another species. We test three key hypotheses: (1) higher-order sensitivity in humans reflects a generic mechanism present across species for analyzing complex sounds like speech and music; (2) higher-order sensitivity reflects an adaptation to ecologically relevant sounds such as speech and music in humans or vocalizations in other species; and (3) higher-order sensitivity reflects a specific adaptation in humans, potentially driven by the unique demands of speech and music perception, that is not generically present in other species even for ecologically relevant sounds. We addressed this question by measuring cortical responses in ferrets – one of the most common animal models used to study auditory cortex (*Nelken et al., 2008*) – to the same set of natural and synthetic sounds previously tested in humans, as well as natural and synthetic ferret vocalizations. Responses were measured using functional ultrasound imaging (fUS) (*Macé et al., 2011*; *Bimbard et al., 2018*), a recently developed wide-field imaging technique that like fMRI detects changes in neural activity via changes in blood flow (movement of blood induces a Doppler effect detectable with ultrasound). fUS has substantially better spatial resolution than fMRI, making it applicable to small animals like ferrets. We found that tuning for spectrotemporal modulations present in both natural and synthetic sounds was similar between humans and animals, and could be quantitatively predicted across species, consistent with prior findings (*Mesgarani et al., 2008*; *Erb et al., 2019*). But unlike humans, ferret responses to natural and synthetic sounds were similar throughout primary and non-primary auditory cortex even

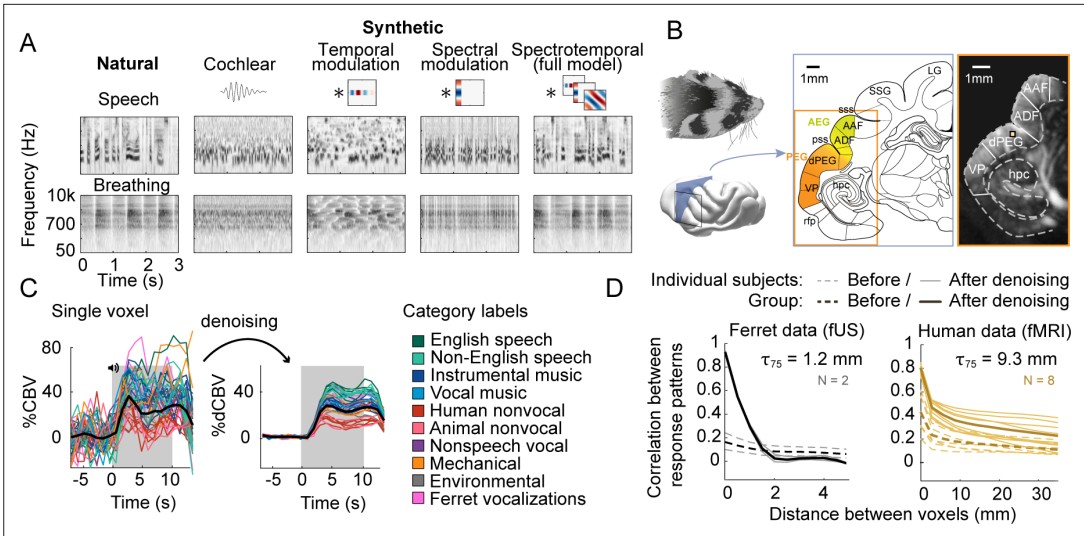

**Figure 1.** Schematic of stimuli and imaging protocol. (**A**) Cochleagrams for two example natural sounds (left column) and corresponding synthetic sounds (right four columns) that were matched to the natural sounds along a set of acoustic statistics of increasing complexity. Statistics were measured by filtering a cochleagram with filters tuned to temporal, spectral, or joint spectrotemporal modulations. (**B**) Schematic of the imaging procedure. A three-dimensional volume, covering all of ferret auditory cortex, was acquired through successive coronal slices. Auditory cortical regions (colored regions) were mapped with anatomical and functional markers (**Radtke-Schuller, 2018**). The rightmost image shows a single ultrasound image with overlaid region boundaries. Auditory regions: dPEG: dorsal posterior ectosylvian gyrus; AEG: anterior ectosylvian gyrus; VP: ventral posterior auditory field; ADF: anterior dorsal field; AAF: anterior auditory field. Non-auditory regions: hpc: hippocampus; SSG: suprasylvian gyrus; LG: lateral gyrus. Anatomical markers: pss: posterior sylvian sulcus; sss: superior sylvian sulcus. (**C**) Response timecourse of a single voxel to all natural sounds, before (left) and after (right) denoising. Each line reflects a different sound, and its color indicates its membership in one of 10 different categories. English and non-English speech are separated out because all of the human subjects tested in our prior study were native English speakers, and so the distinction is meaningful in humans. The gray region shows the time window when sound was present. We summarized the response of each voxel by measuring its average response to each sound between 3 and 11 s post-stimulus onset. The location of this voxel corresponds to the highlighted voxel in panel B. (**D**) We measured the correlation across sounds between pairs of voxels as a function or their distance using two independent measurements of the response (odd vs. even repetitions). Results are plotted separately for ferret fUS data (left) and human fMRI data (right). The 0 mm datapoint provides a measure of test–retest reliability and the fall-off with distance provides a measure of spatial precision. Results are shown before and after component denoising. Note that in our prior fMRI study we did not use component denoising because the voxels were sufficiently reliable; we used component-denoised human data here to make the human and ferret analyses more similar (findings did not depend on this choice: see **Figure 1—figure supplement 2**). The distance needed for the correlation to decay by 75% is shown above each plot ($\tau_{75}$). The human data were smoothed using a 5 MM FWHM kernel, the same amount used in our prior study, but fMRI responses were still coarser when using unsmoothed data ($\tau_{75}$ = 6.5 mm; findings did not depend on the presence/absence of smoothing). Thin lines show data from individual human (N = 8) and ferret (N = 2) subjects, and thick lines show the average across subjects.

The online version of this article includes the following figure supplement(s) for figure 1:

**Figure supplement 1.** The effect of enhancing reliable signal using a procedure similar to 'denoising source separation (DSS)' (see 'Denoising part II' in Materials and methods) (**de Cheveigné and Parra, 2014**).

**Figure supplement 2.** Effect of component denoising on human fMRI results.

when comparing natural and synthetic ferret vocalizations; and the small differences that were present in ferrets were weak and spatially scattered. This finding suggests that representations of natural sounds in humans and ferrets diverge substantially at the final stages of acoustic processing.

## Results

### Experiment I: Comparing ferret cortical responses to natural vs. synthetic sounds

We measured cortical responses with fUS to the same 36 natural sounds tested previously in humans plus four additional ferret vocalizations (experiment II tested many more ferret vocalizations). The 36 natural sounds included speech, music, and other environmental sounds (see *Supplementary file 1*). For each natural sound, we synthesized four sounds that were matched on acoustic statistics of increasing complexity (*Figure 1A*): (1) cochlear energy statistics, (2) temporal modulation statistics, (3) spectral modulation statistics, and (4) spectrotemporal modulation statistics. Cochlear-matched sounds had a similar frequency spectrum, but their modulation content was unconstrained and thus differed from the natural sounds. Modulation-matched sounds were additionally constrained in their temporal and/or spectral modulation rates, measured by linearly filtering a cochleagram representation with filters tuned to different modulation rates (modulation-matched sounds also had matched cochlear statistics so as to isolate the contribution of modulation sensitivity). The modulation-matched sounds audibly differ from their natural counterparts, particularly for complex sounds like speech and music that contain higher-order structure not captured by frequency and modulation statistics (listen to example sounds here). We focused on time-averaged statistics because the hemodynamic response measured by both fMRI and fUS reflects a time-averaged measure of neural activity. As a consequence, each of the synthetic sounds can be thought of as being matched under a different model of the hemodynamic response (*Norman-Haignere et al., 2018*).

We measured fUS responses throughout primary and non-primary ferret auditory cortex (*Figure 1B*). We first plot the response timecourse to all 40 natural sounds for one example voxel in non-primary auditory cortex (dPEG) (*Figure 1C*). We plot the original timecourse of the voxel as well as a denoised version, computed by projecting the timecourse onto a small number of reliable components (see Materials and methods). Our denoising procedure substantially boosted the SNR of the measurements (*Figure 1—figure supplement 1*) and made it possible to analyze individual voxels, as opposed to averaging responses across a large region of interest (ROI), which could potentially wash out heterogeneity present at the single-voxel level. As expected and similar to fMRI, we observed a gradual build-up of the hemodynamic response after stimulus onset. The shape of the response timecourse was similar across stimuli, but the magnitude of the response varied. We thus summarized the response of each voxel to each sound by its time-averaged response magnitude (the same approach used in our prior fMRI study). We found that the denoised fUS responses were substantially more reliable and precise than the fMRI voxels from our prior study (*Figure 1D*) (test–retest correlation: 0.93 vs. 0.44, Wilcoxon rank-sum test across subjects, p<0.01). To make our human and ferret analyses more similar, we used component-denoised fMRI data in this study, which had similar reliability to the denoised fUS data (*Figure 1D*; results were similar without denoising, see *Figure 1—figure supplement 2*).

We next plot the response of two example fUS voxels – one in primary auditory cortex (A1) and one in a non-primary area (dPEG) – to natural and corresponding synthetic sounds that have been matched on the full spectrotemporal modulation model (*Figure 2A*; results were similar when averaging responses within anatomical regions of interest, see *Figure 2—figure supplement 1*). For comparison, we plot the test–retest reliability of each voxel across repeated presentations of the same sound (*Figure 2B*), as well as corresponding figures from two example voxels in human primary/non-primary auditory cortex (*Figure 2C and D*). As in our prior study, we quantified the similarity of responses to natural and synthetic sounds using the normalized squared error (NSE). The NSE takes a value of 0 if responses to natural and synthetic sounds are the same, and 1 if there is no correspondence between the two (see Materials and methods for details).

Both the primary and non-primary ferret voxels produced similar responses to natural and corresponding synthetic sounds (NSEs: 0.084, 0.13), suggesting that spectrotemporal modulations are sufficient to account for most of the response variance in these voxels. The human primary voxel also

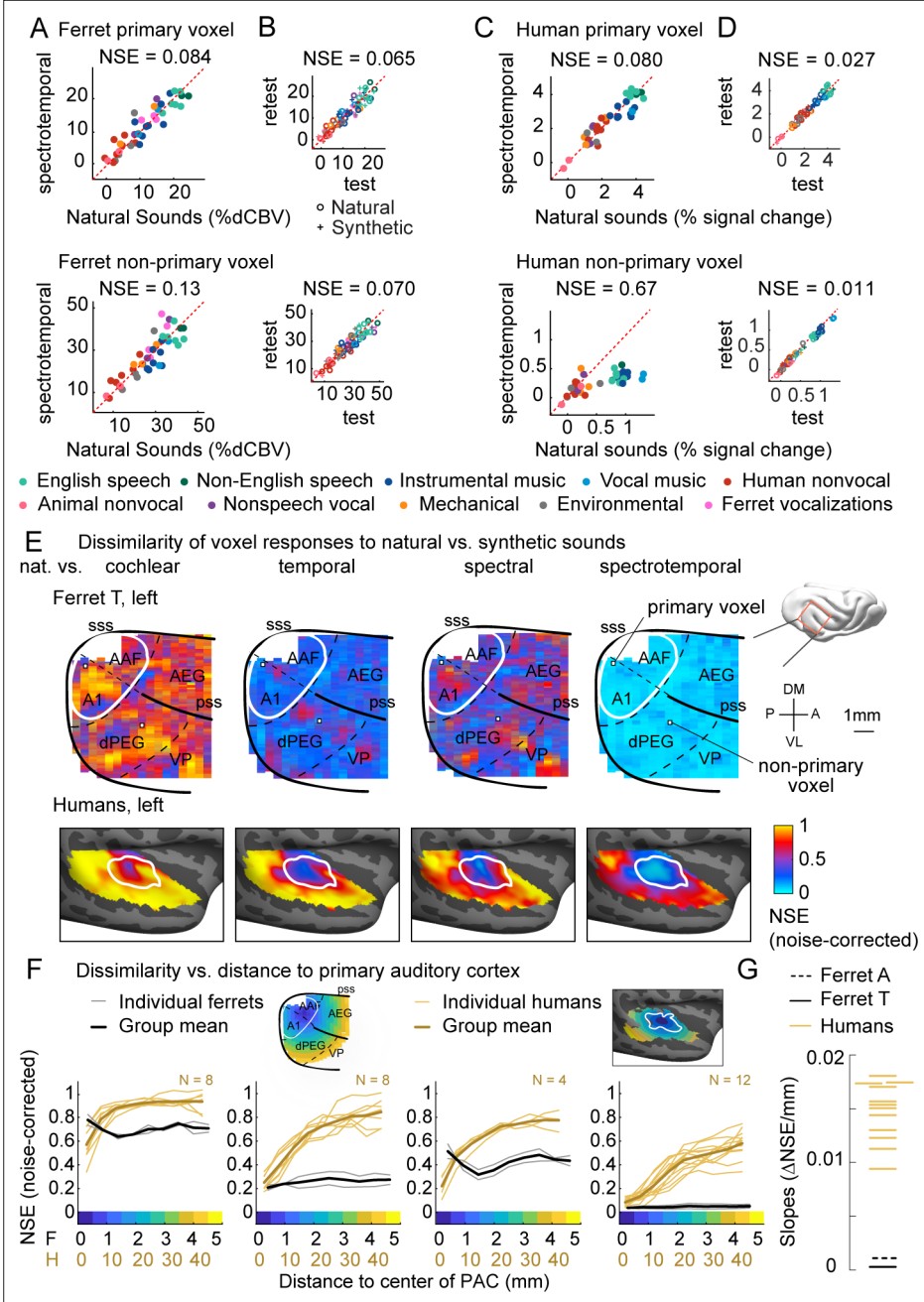

**Figure 2.** Dissimilarity of responses to natural vs. synthetic sounds in ferrets and humans. (**A**) Response of two example fUS voxels to natural and corresponding synthetic sounds with matched spectrotemporal modulation statistics. Each dot shows the time-averaged response to a single pair of natural/synthetic sounds (after denoising), with colors indicating the sound category. The example voxels come from primary (top, A1) and non-primary (bottom, dPEG) regions of the ferret auditory cortex (locations shown in panel E). The normalized squared error (NSE) quantifies the dissimilarity of responses. (**B**) Test–retest response of the example voxels across all natural (o) and synthetic (+) sounds (odd vs. even repetitions). The responses were highly reliable due to the denoising procedure. (**C, D**) Same as panels (**A, B**), but showing two example voxels from human primary/non-primary auditory cortex. (**E**) Maps plotting the dissimilarity of responses to natural vs. synthetic sounds from one ferret hemisphere (top row) and from humans (bottom row). Each column shows results for a different set of synthetic sounds. The synthetic sounds were constrained by statistics of increasing complexity (from left to right): just cochlear statistics, cochlear + temporal modulation statistics, cochlear + spectral modulation statistics, and cochlear + spectrotemporal modulation statistics. Dissimilarity was quantified using the NSE, corrected for noise using the test–retest reliability of the voxel responses. Ferret maps show a 'surface' view from above of the sylvian

*Figure 2 continued on next page*

*Figure 2 continued*

gyri, similar to the map in humans. Surface views were computed by averaging activity perpendicular to the cortical surface. The border between primary and non-primary auditory cortex is shown with a white line in both species and was defined using tonotopic gradients. Areal boundaries in the ferret are also shown (dashed thin lines). This panel shows results from one hemisphere of one animal (ferret T, left hemisphere), but results were similar in other animals/hemispheres (*Figure 2—figure supplement 2*). The human map is a group map averaged across 12 subjects, but results were similar in individual subjects (*Norman-Haignere et al., 2018*). (**F**) Voxels were binned based on their distance to primary auditory cortex (defined tonotopically). This figure plots the median NSE value in each bin. Each thin line corresponds to a single ferret (gray) or a single human subject (gold). Thick lines show the average across all subjects. The ferret and human data were rescaled so that they could be plotted on the same figure, using a scaling factor of 10, which roughly corresponds to the difference in the radius of primary auditory cortex between ferrets and humans. The corresponding unit is plotted on the x-axis below. The number of human subjects varied by condition (see Materials and methods for details) and is indicated on each plot. (**G**) The slope of NSE vs. distance-to-primary auditory cortex (PAC) curve (**F**) from individual ferret and human subjects using responses to the spectrotemporally matched synthetic sounds. We used absolute distances to quantify the slope, which is conservative with respect to the hypothesis since correcting for brain size would differentially increase the ferret slopes.

The online version of this article includes the following figure supplement(s) for figure 2:

**Figure supplement 1.** Responses to natural and synthetic sounds in standard anatomical regions of interest (ROIs).

**Figure supplement 2.** Dissimilarity maps for all hemispheres and animals.

**Figure supplement 3.** Uncorrected normalized squared error (NSE) values.

---

showed similar responses to natural and synthetic responses (NSE: 0.080). In contrast, the human non-primary voxel responded substantially more to the natural speech (green) and music (blue) than matched synthetics, yielding a high NSE value (0.67). This pattern demonstrates that spectrotemporal modulations are insufficient to drive the response of the human non-primary voxel, plausibly because it responds to higher-order features that are not captured by modulation statistics.

We quantified this trend across voxels by plotting maps of the NSE between natural and synthetic sounds (*Figure 2E* shows one hemisphere of one animal, but results were very similar in other hemispheres of other animals, see *Figure 2—figure supplement 2*). We used the test–retest reliability of the responses to noise-correct the measured NSE values such that the effective noise floor given the reliability of the measurements is zero. We show separate maps for each of the different sets of statistics used to constrain the synthetic sounds (cochlear, temporal modulation, spectral modulation, and spectrotemporal modulation). Each map shows a view from above auditory cortex, computed by averaging NSE values perpendicular to the cortical sheet. We summarized the data in this way because we found that maps were similar across the different layers within a cortical column. Below we plot corresponding maps from humans. The human maps are based on data averaged across subjects, but similar results were observed in individual subjects (*Norman-Haignere et al., 2018*).

In ferrets, we found that responses became more similar as we matched additional acoustic features, as expected (NSE spectrotemporal < NSE temporal < NSE spectral < NSE cochlear, p<0.01 in every ferret; significance computed via bootstrapping across sounds the median NSE value across all voxels in auditory cortex). Notably, we observed similar NSE values in primary and non-primary regions for all conditions, and for sounds matched on joint spectrotemporal statistics, NSE values were close to 0 throughout most of auditory cortex. This pattern contrasts sharply with that observed in humans, where we observed a clear and substantial rise in NSE values when moving from primary to non-primary regions even for sounds matched on joint spectrotemporal modulations statistics. We quantified these effects by binning voxels based on their distance to primary auditory cortex, as was done previously in humans (*Figure 2F*; see *Figure 2—figure supplement 3* for results without noise correction), and then measuring the slope of the NSE-vs.-distance curve for each human subject and each ferret tested (*Figure 2G*). We used absolute distances for calculating the slopes, which is a highly conservative choice given our findings since correcting for brain size would enhance the slopes of ferrets relative to humans. Despite this choice, we found that the slope of every ferret was well below that of all 12 human subjects tested, and thus significantly different from the human group via a non-parametric sign test (p<0.001). This finding demonstrates that the higher-order sensitivity we

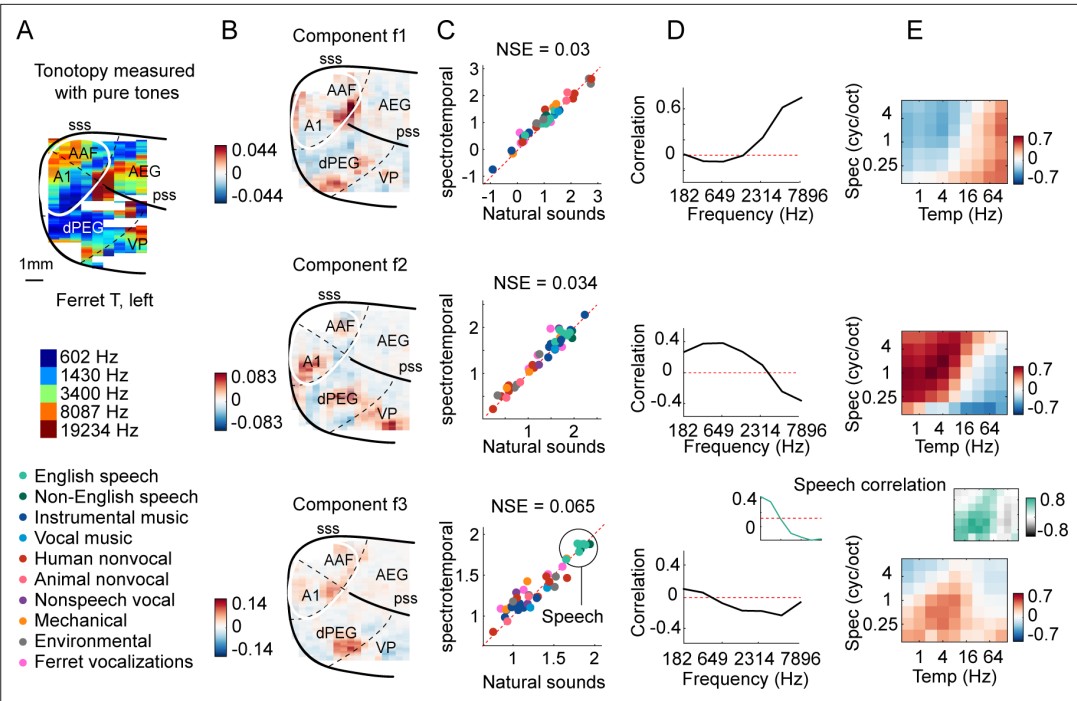

**Figure 3.** Organization of frequency and modulation tuning in ferret auditory cortex, as revealed by component analysis. (**A**) For reference with the weight maps in panel (**B**), a tonotopic map is shown, measured using pure tones. The map is from one hemisphere of one animal (ferret T, left). (**B**) Voxel weight maps from three components, inferred using responses to natural and synthetic sounds (see *Figure 3—figure supplement 1* for all eight components and *Figure 3—figure supplement 2* for all hemispheres). The maps for components f1 and f2 closely mirrored the high- and low-frequency tonotopic gradients, respectively. (**C**) Component response to natural and spectrotemporally matched synthetic sounds, colored based on category labels (labels shown at the bottom left of the figure). Component f3 responded preferentially to speech sounds. (**D**) Correlation of component responses with energy at different audio frequencies, measured from a cochleagram. Inset for f3 shows the correlation pattern that would be expected from a response that was perfectly speech selective (i.e., 1 for speech, 0 for all other sounds). (**E**) Correlations with modulation energy at different temporal and spectral rates. Inset shows the correlation pattern that would be expected for a speech-selective response. Results suggest that f3 responds to particular frequency and modulation statistics that happen to differ between speech and other sounds.

The online version of this article includes the following figure supplement(s) for figure 3:

**Figure supplement 1.** Results from all eight ferret components.

**Figure supplement 2.** Component weight maps from all hemispheres and ferrets.

**Figure supplement 3.** Human components.

**Figure supplement 4.** Predicting human component responses from ferret components.

**Figure supplement 5.** Predicting ferret component responses from human components.

previously observed for natural sounds in human non-primary auditory cortex is not a generic feature of higher-order processing in mammals.

## Assessing and comparing sensitivity for frequency and modulation across species

Our NSE maps suggest that ferret cortical responses are tuned for frequency and modulation, but do not reveal how this tuning is organized or whether it is similar to that in humans. While it is not feasible to inspect or plot all individual voxels, we found that fUS responses like human fMRI responses are low-dimensional and can be explained as the weighted sum of a small number of component response patterns. This observation served as the basis for our denoising procedure, as well as a useful way to examine ferret cortical responses and to compare those responses with humans. We found that we

could discriminate approximately eight distinct component response patterns before overfitting to noise.

We first examined the inferred response patterns and their anatomical distribution of weights in the brain (*Figure 3* shows three example components; *Figure 3—figure supplement 1* shows all eight components). All of the component response profiles showed significant correlations with measures of energy at different cochlear frequencies and spectrotemporal modulation rates (*Figure 3D and E*) (p<0.01 for all components for both frequency and modulation features; statistics computed via a permutation test across the sound set). Two components (f1 and f2) had responses that correlated with energy at high and low frequencies, respectively, with voxel weights that mirrored the tonotopic gradients measured in these animals (compare *Figure 3A* with *Figure 3B*; see *Figure 3—figure supplement 2* for all hemispheres/animals), similar to the tonotopic components previously identified in humans (*Norman-Haignere et al., 2015*; *Figure 3—figure supplement 3*, components h1 and h2). We also observed components with weak frequency tuning but prominent tuning for spectrotemporal modulations (*Figure 3—figure supplement 1*), again similar to humans. Perhaps surprisingly, one component (f3) responded preferentially to speech sounds, and its response correlated with energy at frequency and modulation rates characteristic of speech (insets in *Figure 3D and E*, bottom row). But notably, all of the inferred components, including the speech-preferring component, produced very similar responses to natural and synthetic sounds (*Figure 3C*), suggesting that their response can be explained by tuning for frequency and modulation. This contrasts with the speech- and music-preferring components previously observed in humans, which showed a clear response preference for natural speech and music, respectively, and which clustered in distinct non-primary regions of human auditory cortex (see *Figure 3—figure supplement 3*, components h5 and h6). This finding shows that preferential responses for natural speech compared with other natural sounds are not unique to humans, and thus that comparing responses to natural vs. synthetic sounds is critical to revealing representational differences between species.

Overall, the frequency and modulation tuning evident in the ferret components appeared similar to that in humans (*Norman-Haignere et al., 2015*). To quantitatively evaluate similarity, we attempted to predict the response of each human component, inferred from our prior work, from those in the ferrets (*Figure 3—figure supplement 4*) and vice versa (*Figure 3—figure supplement 5*). We found that much of the component response variation to synthetic sounds could be predicted across species (*Figure 3—figure supplement 4B,D and E*, *Figure 3—figure supplement 5A, C and D*). This finding is consistent with the hypothesis that tuning for frequency and modulation is similar across species since the synthetic sounds only varied in their frequency and modulation statistics. In contrast, differences between natural vs. synthetic sounds were only robust in humans and as a consequence could not be predicted from responses in ferrets (*Figure 3—figure supplement 4C, D and E*). Thus, frequency and modulation tuning are both qualitatively and quantitatively similar across species, despite substantial differences in higher-order sensitivity.

## Experiment II: Testing the importance of ecological relevance

The results of experiment I show that higher-order sensitivity in humans is not a generic feature of auditory processing for complex sounds. However, the results could still be explained by a difference in ecological relevance since differences between natural and synthetic sounds in humans are mostly driven by speech and music (*Norman-Haignere et al., 2018*) and experiment I included more speech (8) and music (10) sounds than ferret vocalizations (4). To test this possibility, we performed a second experiment that included many more ferret vocalizations (30), as well as a smaller number of speech (14) and music (16) sounds to allow comparison with experiment I. We only synthesized sounds matched in their full spectrotemporal modulation statistics to be able to test a broader sound set.

Despite testing many more ferret vocalizations, results were nonetheless similar to those of experiment I: voxel responses to natural and synthetic sounds were similar throughout primary and non-primary auditory cortex, yielding low NSE values everywhere (*Figure 4A*). We also observed similar component responses to those observed in experiment I (*Figure 4—figure supplement 2*). To directly test if ferrets showed preferential responses to natural vs. synthetic ferret vocalizations, we computed maps plotting the average difference between natural vs. synthetic sounds for different categories, using data from both experiments I and II (*Figure 4B*). We also separately measured the NSE for sounds from different categories, again plotting NSE values as a function of distance to PAC (*Figure 4C and*

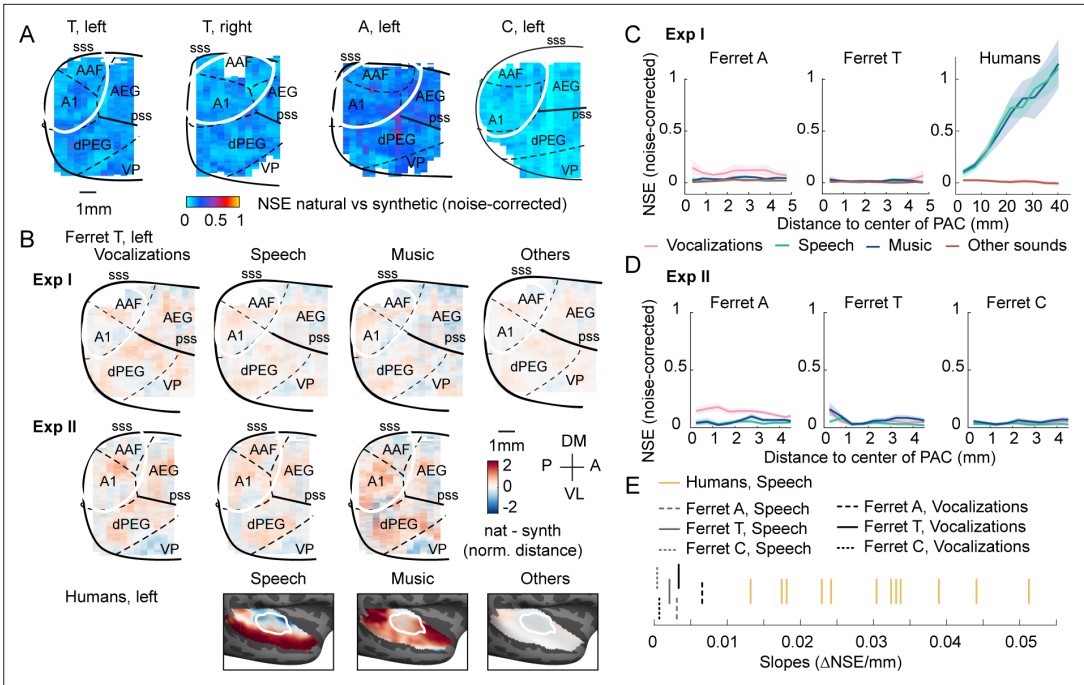

**Figure 4.** Testing the importance of ecological relevance. Experiment II measured responses to a larger number of ferret vocalizations (30 compared with 4 in experiment I), as well as speech (14) and music (16) sounds. (**A**) Map showing the dissimilarity between natural and spectrotemporally matched synthetic sounds from experiment II for each recorded hemisphere, measured using the noise-corrected normalized squared error (NSE). NSE values were low across auditory cortex, replicating experiment I. (**B**) Maps showing the average difference between responses to natural and synthetic sounds for vocalizations, speech, music, and others sounds, normalized for each voxel by the standard deviation across all sounds. Results are shown for ferret T, left hemisphere for both experiments I and II (see **Figure 4—figure supplement 1C** for all hemispheres). For comparison, the same difference maps are shown for the human subjects, who were only tested in experiment I. (**C**) NSE for different sound categories, plotted as a function of distance to primary auditory cortex (binned as in **Figure 2F**). Shaded area represents 1 standard error of the mean across sounds within each category (**Figure 4—figure supplement 1D** plots NSEs for individual sounds). (**D**) Same as panel (**C**) but showing results from experiment II. (**E**) The slope of NSE vs. distance-to-primary auditory cortex (PAC) curves for individual ferrets and human subjects. Ferret slopes were measured separately for ferret vocalizations (black lines) and speech (gray lines) (animal indicated by line style). For comparison, human slopes are plotted for speech (each yellow line corresponds to a different human subject).

The online version of this article includes the following figure supplement(s) for figure 4:

**Figure supplement 1.** Results of experiment II from other hemispheres.

**Figure supplement 2.** Components from experiment II.

**Figure supplement 3.** The effect of removing outside-of-cortex components on motion correlations.

---

*D*). The differences that we observed between natural and synthetic sounds were small and scattered throughout primary and non-primary auditory cortex, even for ferret vocalizations. In one animal, we observed significantly larger NSE values for ferret vocalizations compared with speech and music (ferret A, $Md_{voc}$ = 0.1 4 vs. $Md_{SpM}$ = 0.042, Wilcoxon rank-sum test: T = 1138, z = 3.29, p<0.01). But this difference was not present in the other two ferrets tested (p>0.55) and was also not present when we averaged NSE values across animals ($Md_{voc}$ = 0.053 vs. $Md_{SpM}$ = 0.033, Wilcoxon rank-sum test: T = 1016, z = 1.49, p=0.27). Moreover, the slope of the NSE vs. distance-to-PAC curve was near 0 for all animals and sound categories, even for ferret vocalizations, and was substantially lower than the slopes measured in all 12 human subjects (**Figure 4E**) (vocalizations in ferrets vs. speech in humans: p<0.001 via a sign test; speech in ferrets vs. speech in humans: p<0.001). In contrast, human cortical responses were substantially larger for natural vs. synthetic speech and music, and these response enhancements were concentrated in distinct non-primary regions (lateral for speech and anterior/ posterior for music) and different from those for other natural sounds (**Figure 4B**). Thus, ferrets do not

show any of the neural signatures of higher-order sensitivity that we previously identified in humans (large effect size, spatially clustered responses, and a clear non-primary bias), even for conspecific vocalizations.

Given the weak neural differences between natural and synthetic sounds, we wondered if any of the animals could perceive the difference between natural and synthetic sounds. Using a video recording of the animals' face (*Figure 4—figure supplement 1A*), we found that one ferret (ferret A) spontaneously moved more during the presentation of the natural ferret vocalizations compared with both the synthetic sounds (*Figure 4—figure supplement 1B*, $Md_{voc, nat}$ = 1.77 vs. $Md_{voc, synth}$ = 1.07, Wilcoxon signed-rank test across sounds: T = 464, z = 4.76, p<0.001) and the other natural sounds ($Md_{voc, nat}$ = 1.8 vs. $Md_{others, nat}$ = 0.65, Wilcoxon rank-sum test across sounds T = 1301, z = 5.70, p<0.001). There was a similar trend in a second animal (ferret T; $Md_{voc, nat}$ = 1.68 vs. $Md_{voc, synth}$ = 1.44, T = 335, z = 2.11, p=0.07; $Md_{voc, nat}$ = 1.6 vs. $Md_{others, nat}$ = 0.97, T = 1269, z = 5.23, p<0.001), but not in the third (ferret C; $Md_{voc, nat}$ = 0.41 vs. $Md_{voc, synth}$ = 0.47, T = 202, z = –0.62, p=0.53), likely because the animal did not move very much for any of the sounds. This finding demonstrates that ferrets are perceptually capable of detecting the difference between natural and synthetic sounds without any overt training and that this difference is more salient for ferret vocalizations, consistent with their greater ecological relevance. Since our key neural findings were present in all animals tested, including ferret A, we conclude that our results cannot be explained by an inability to perceptually detect differences between natural and synthetic vocalizations.

## Discussion

Our study reveals a prominent divergence in the representation of natural sounds between humans and ferrets. Using a recently developed wide-field imaging technique (fUS), we measured cortical responses in the ferret to a set of natural and spectrotemporally matched synthetic sounds previously tested in humans. We found that tuning for frequency and modulation statistics in the synthetic sounds was similar across species. But unlike humans, who showed preferential responses to natural vs. synthetic speech and music in non-primary regions, ferret cortical responses to natural and synthetic sounds were similar throughout primary and non-primary auditory cortex, even when tested with ferret vocalizations. This finding suggests that higher-order sensitivity in humans for natural vs. synthetic speech/music (1) does not reflect a species-generic mechanism for analyzing complex sounds and (2) does not reflect a species-generic adaptation for coding ecologically relevant sounds like conspecific vocalizations. Instead, our findings suggest that auditory representations in humans diverge from ferrets at higher-order processing stages, plausibly driven by the unique demands of speech and music perception.

### Species differences in the representation of natural sounds

The central challenge of sensory coding is that behaviorally relevant information is often not explicit in the inputs to sensory systems. As a consequence, sensory systems transform their inputs into higher-order representations that expose behaviorally relevant properties of stimuli (*DiCarlo and Cox, 2007*; *Mizrahi et al., 2014*; *Theunissen and Elie, 2014*). The early stages of this transformation are thought to be conserved across many species. For example, all mammals transduce sound pressure waveforms into a frequency-specific representation of sound energy in the cochlea, although the resolution and frequency range of cochlear tuning differ across species (*Bruns and Schmieszek, 1980*; *Koppl et al., 1993*; *Joris et al., 2011*; *Walker et al., 2019*). But it has remained unclear whether representations at later stages are similarly conserved across species.

Only a few studies have attempted to compare cortical representations of natural sounds between humans and other animals, and these studies have typically found similar representations in auditory cortex. Studies of speech phonemes in ferrets (*Mesgarani et al., 2008*) and macaques (*Steinschneider et al., 2013*) have replicated neural phenomena observed in humans (*Mesgarani et al., 2014*). A recent fMRI study found that maps of spectrotemporal modulation tuning, measured using natural sounds, are coarsely similar between humans and macaques, although slow temporal modulations that are prominent in speech were better decoded in humans compared with macaques (*Erb et al., 2019*), potentially analogous to prior findings of enhanced cochlear frequency tuning for behaviorally relevant sound frequencies (*Bruns and Schmieszek, 1980*; *Koppl et al., 1993*). Thus, prior

work has revealed differences in the extent and resolution of neural tuning for different acoustic frequencies and modulation rates.

Our study demonstrates that human non-primary regions exhibit a form of higher-order acoustic sensitivity that is almost completely absent in ferrets. Ferret cortical responses to natural and spectrotemporally matched synthetic sounds were closely matched throughout their auditory cortex, and the small differences that we observed were scattered throughout primary and non-primary regions (*Figure 4B*), unlike the pattern observed in humans. As a consequence, the differences that we observed between natural and synthetic sounds in humans were not predictable from cortical responses in ferrets, even though we could predict responses to synthetic sounds across species (*Figure 3—figure supplement 4*). This higher-order sensitivity is unlikely to be explained by explicit semantic knowledge about speech or music since similar responses are observed for foreign speech (*Norman-Haignere et al., 2015*; *Norman-Haignere et al., 2018*) and music sensitivity is robust in listeners without musical training (*Boebinger et al., 2020*). These results suggest that humans develop or have evolved a higher-order stage of acoustic analysis, potentially specific to speech and music, that cannot be explained by standard frequency and modulation statistics and is largely absent from the ferret brain. This specificity for speech and music could be due to their acoustic complexity, their behavioral relevance to humans, or a combination of the two.

By comparison, our study suggests that there is a substantial amount of cross-species overlap in the cortical representation of frequency and modulation features. Both humans and ferrets exhibited tonotopically organized tuning for different audio frequencies. Like humans, ferrets showed spatially organized sensitivity for different temporal and spectral modulation rates that coarsely mimicked the types of tuning we have previously observed in humans, replicating prior findings (*Erb et al., 2019*). And this tuning was sufficiently similar that we could quantitatively predict response patterns to the synthetic sounds across species (*Figure 3—figure supplement 4*). These results do not imply that frequency and modulation tuning is the same across species, but do suggest that the organization is similar.

Our results also do not imply that ferrets lack higher-order acoustic representations. Indeed, we found that one ferret's spontaneous movements robustly discriminated between natural and synthetic ferret vocalizations, demonstrating behavioral sensitivity to the features that distinguish these sound sets. But how species-relevant higher-order features are represented is likely distinct between humans and ferrets. Consistent with this idea, we found that differences between natural and synthetic sounds are weak, distributed throughout primary and non-primary regions, and show a mix of enhanced and suppressive responses (*Figure 4C*), unlike the strong response enhancements we observed for natural speech and music in distinct regions of human non-primary auditory cortex.

The species differences we observed are unlikely to be driven by differences in the method used to record brain responses (fUS vs. fMRI) for several reasons. First, both methods detect changes in neural responses driven by hemodynamic activity. Second, the denoised fUS responses were both more reliable and more spatially precise than our previously analyzed fMRI voxels. Higher SNR and spatial precision should make it easier, not harder, to detect response differences between sounds, like the natural and synthetic sounds tested here. Third, all of our measures were noise-corrected and thus any residual differences in SNR between species or brain regions should have minimal effect on our measures. Fourth, human non-primary regions show a strong response preference for natural vs. synthetic sounds that is absent in ferrets, and there is no reason why methodological differences should produce a greater response to one set of sounds over another in a specific anatomical region of one species. Fifth, ferrets' cortical responses show clear selectivity for standard frequency and modulation features of sound, and this selectivity is qualitatively similar to that observed in humans. Sixth, the differences we observed between humans and ferrets are not subtle: humans show a substantial change across their auditory cortex in sensitivity for natural vs. synthetic sounds while ferrets show no detectable change across their auditory cortex. We quantified this change by measuring the slope of the NSE-vs.-distance curve and found that the slopes in ferrets were close to zero and differed substantially from every human subject tested.

A recent study also found evidence for a species difference in auditory cortical organization by comparing responses to tone and noise stimuli between humans and macaques (*Norman-Haignere et al., 2019*). This study found that preferential responses to tones vs. noise were larger in both primary and non-primary regions of human auditory cortex compared with macaques, which might

reflect the importance of speech and music in humans where harmonic structure plays a central role. Our findings are unlikely to reflect greater tone sensitivity in humans because the differences that we observed between natural and synthetic sounds were not limited to tone-selective regions. Here, we tested a mucher wider range of natural and synthetic sounds that differ on many different ecologically relevant dimensions and we could thus compare the overall functional organization between humans and ferrets. As a consequence, we were able to identify a substantial divergence in neural representations at a specific point in the cortical hierarchy.

## Methodological advances

Our findings were enabled by a recently developed synthesis method that makes it possible to synthesize sounds with frequency and modulation statistics that are closely matched to those in natural sounds (*Norman-Haignere et al., 2018*). Because the synthetics are otherwise unconstrained, they lack higher-order acoustic properties present in complex natural sounds like speech and music (e.g., syllabic structure; musical notes, harmonies, and rhythms). Comparing neural responses to natural and synthetic sounds thus provides a way to isolate responses to higher-order properties of natural stimuli that cannot be accounted for by modulation statistics. This methodological advance was critical to differentiating human and ferret cortical responses. Indeed, when considering natural or synthetic sounds alone, we observed similar responses between species. We even observed preferential responses to speech compared with other natural sounds in the ferret auditory cortex due to the fact that speech has a unique range of spectrotemporal modulations. Thus, if we had only tested natural sounds, we might have concluded that speech-sensitive responses in the human non-primary auditory cortex reflect the same types of acoustic representations present in ferrets.

Our study illustrates the utility of wide-field imaging methods in comparing the brain organization of different species (*Bimbard et al., 2018*; *Milham et al., 2018*). Most animal physiology studies focus on measuring responses from single neurons or small clusters of neurons in a single brain region. While this approach is essential to understanding the neural code at a fine grain, studying a single brain region can obscure larger-scale trends that are evident across the cortex. Indeed, if we had only measured responses in a single region of auditory cortex, we would have missed the most striking difference between humans and ferrets: the emergence of preferential responses to natural sounds in non-primary regions of humans but not ferrets (*Figure 2E*).

fUS imaging provides a powerful way of studying large-scale functional organization in small animals such as ferrets since it has better spatial resolution than fMRI (*Macé et al., 2011*; *Bimbard et al., 2018*). Because fUS responses are noisy, prior studies, including those from our lab, have only been able to characterize responses to a single stimulus dimension, such as frequency, typically using a small stimulus set (*Gesnik et al., 2017*; *Bimbard et al., 2018*). Here, we developed a denoising method that made it possible to measure highly reliable responses to over a hundred stimuli in a single experiment. We were able to recover at least as many response dimensions as those detectable with fMRI in humans, and those response dimensions exhibited sensitivity for a wide range of frequencies and modulation rates. Our study thus pushes the limits of what is possible using ultrasound imaging and establishes fUS as an ideal method for studying the large-scale functional organization of the animal brain.

## Assumptions and limitations

The natural and synthetic sounds we tested were closely matched in their time-averaged cochlear frequency and modulation statistics, measured using a standard model of cochlear and cortical modulation tuning (*Chi et al., 2005*; *Norman-Haignere et al., 2018*). We focused on time-averaged statistics because fMRI and fUS reflect time-averaged measures of neural activity due to the temporally slow nature of hemodynamic responses. Thus, a similar response to natural and synthetic sounds indicates that the statistics being matched are sufficient to explain the voxel response. By contrast, a divergent voxel response indicates that the voxel responds to features of sound that are not captured by the model.

While divergent responses by themselves do not demonstrate a higher-order response, there are several reasons to think that the sensitivity we observed in human non-primary regions is due to higher-order tuning. First, the fact that differences between natural and synthetic speech/music were much larger in non-primary regions suggests that these differences are driven by higher-order

processing above and beyond that present in primary auditory cortex, where spectrotemporal modulations appear to explain much of the voxel response. Second, the natural and synthetic sounds produced by our synthesis procedure are in practice closely matched on a wide variety of spectrotemporal filterbank models (*Norman-Haignere et al., 2018*). As a consequence, highly divergent responses to natural and synthetic sounds rule out many such models. Third, the fact that responses were consistently larger for natural speech/music vs. synthetic speech/music suggests that these non-primary regions respond preferentially to features in natural sounds that are not explicitly captured by spectrotemporal modulations and are thus absent from the synthetic sounds.

Our findings show that a prominent signature of hierarchical functional organization present in humans – preferential responses for natural vs. spectrotemporal structure – is largely absent in ferret auditory cortex. But this finding does not imply that there is no functional differentiation between primary and non-primary regions in ferrets. For example, ferret non-primary regions show longer latencies, greater spectral integration bandwidths, and stronger task-modulated responses compared with primary regions (*Elgueda et al., 2019*). The fact that we did not observe differences between primary and non-primary regions is not because the acoustic features manipulated are irrelevant to ferret auditory cortex, since our analysis shows that matching frequency and modulation statistics is sufficient to match the ferret cortical response, at least as measured by ultrasound. Indeed, if anything, it appears that modulation features are more relevant to the ferret auditory cortex since these features appear to drive responses throughout primary and non-primary regions, unlike human auditory cortex where we only observed strong, matched responses in primary regions.

As with any study, our conclusions are limited by the precision and coverage of our neural measurements. For example, fine-grained temporal codes, which have been suggested to play an important role in vocalization coding (*Schnupp et al., 2006*), cannot be detected with fUS. However, we note that the resolution of fUS is substantially better than fMRI, particularly in the spatial dimension and thus the species differences we observed are unlikely to be explained by differences in the resolution of fUS vs. fMRI. It is also possible that ferrets might show more prominent differences between natural and synthetic sounds outside of auditory cortex. But even if this were true, it would still demonstrate a clear species difference because humans show robust sensitivity for natural sounds in non-primary regions just outside of primary auditory cortex, while ferrets apparently do not.

## Possible nature and causes of differences in higher-order sensitivity

What features might non-primary human auditory cortex represent, given that spectrotemporal modulations fail to explain much of the response? Although these regions respond preferentially to speech and music, they are not driven by semantic meaning or explicit musical training (*Overath et al., 2015*; *Boebinger et al., 2020*), are located just beyond primary auditory cortex, and show evidence of having short integration windows on the scale of hundreds of milliseconds (*Overath et al., 2015*; *Norman-Haignere et al., 2020*). This pattern suggests nonlinear sensitivity for short-term temporal and spectral structure present in speech syllables or musical notes (e.g., harmonic structure, pitch contours, and local periodicity). This hypothesis is consistent with recent work showing sensitivity to phonotactics in non-primary regions of the superior temporal gyrus (*Leonard et al., 2015*; *Brodbeck et al., 2018*; *Di Liberto et al., 2019*), and with a recent study showing that deep neural networks trained to perform challenging speech and music tasks are better able to predict responses in non-primary regions of human auditory cortex (*Kell et al., 2018*).

Why don't we observe similar neural sensitivity in ferrets for vocalizations? Ferret vocalizations exhibit additional structure not captured by spectrotemporal modulations since at least one ferret was able to detect the difference between natural and synthetic sounds. However, this additional structure may play a less-essential role in their everyday hearing compared with that of speech and music in humans. Other animals that depend more on higher-order acoustic representations might show more human-like sensitivity in non-primary regions. For example, marmosets have a relatively complex vocal repertoire (*Agamaite et al., 2015*) and depend more heavily on vocalizations than many other species (*Eliades and Miller, 2017*), and thus might exhibit more prominent sensitivity for higher-order properties in their calls. It may also be possible to experimentally enhance sensitivity for higher-order properties via extensive exposure and training, particularly at an early age of development (*Polley et al., 2006*; *Srihasam et al., 2014*). All of these questions could be addressed in future work using the methods developed here.

# Materials and methods

## Animal preparation

Experiments were performed in three head-fixed awake ferrets (A, T, and C), across one or both hemispheres (study 1: $A_{left}$, $A_{right}$, $T_{left}$, $T_{right}$; study 2: $A_{left}$, $T_{left}$, $T_{right}$, $C_{left}$). Ferrets A and C were mothers (had one litter of pups), while ferret T was a virgin. Experiments were approved by the French Ministry of Agriculture (protocol authorization: 21022) and strictly comply with the European directives on the protection of animals used for scientific purposes (2010/63/EU). Animal preparation and fUS imaging were performed as in *Bimbard et al., 2018*. Briefly, a metal headpost was surgically implanted on the skull under anesthesia. After recovery from surgery, a craniotomy was performed over auditory cortex and then sealed with an ultrasound-transparent Polymethylpentene (TPX) cover, embedded in an implant of dental cement. Animals could then recover for 1 week, with unrestricted access to food, water, and environmental enrichment. Imaging windows were maintained across weeks with appropriate interventions when tissue and bone regrowth were shadowing brain areas of interest.

Ultrasound imaging fUS data are collected as a series of 2D images or 'slices.' Slices were collected in the coronal plane and were spaced 0.4 mm apart. The slice plane was varied across sessions to cover the ROI, which included both primary and non-primary regions of auditory cortex. We did not collect data from non-auditory regions due to limited time/coverage. One or two sessions were performed on each day of recording. The resolution of each voxel was 0.1 × 0.1 × ~0.4 mm (the latter dimension, called elevation, being slightly dependent on the depth of the voxel). The overall voxel volume (0.004 mm³) was more than a thousand times smaller than the voxel volume used in our human study (which was either 8 or 17.64 mm³ depending on the subjects/paradigm), which helps to account for their smaller brain.

A separate 'power Doppler' image/slice was acquired every second. Each of these images was computed by first collecting 300 sub-images or 'frames' in a short 600 ms time interval (500 Hz sampling rate). Those 300 frames were then filtered to discard global tissue motion from the signal (*Demené et al., 2015*) (the first 55 principal components (PCs) were discarded because they mainly reflect motion; see *Demené et al., 2015* for details). The blood signal energy, also known as power Doppler, was computed for each voxel by summing the squared magnitudes across the 300 frames separately for each pixel (*Macé et al., 2011*). Power Doppler is approximately proportional to blood volume (*Macé et al., 2011*).

Each of the 300 frames was itself computed from 11 tilted plane wave emissions (–10° to 10° with 2° steps) fired at a pulse repetition frequency of 5500 Hz. Frames were reconstructed from these plane wave emissions using an in-house, GPU-parallelized delay-and-sum beamforming algorithm (*Macé et al., 2011*).

## Stimuli for experiment I

We tested 40 natural sounds: 36 sounds from our prior experiment plus 4 ferret vocalizations (fight call, pup call, fear vocalization, and play call). Each natural sound was 10 s in duration. For each natural sound, we synthesized four synthetic sounds, matched on a different set of acoustic statistics of increasing complexity: cochlear, temporal modulation, spectral modulation, and spectrotemporal modulation. The modulation-matched synthetics were also matched in their cochlear statistics to ensure that differences between cochlear and modulation-matched sounds must be due to the addition of modulation statistics. The natural and synthetic sounds were identical to those in our prior paper, except for the four additional ferret vocalizations, which were synthesized using the same algorithm. We briefly review the algorithm below.

Cochlear statistics were measured from a cochleagram representation of sound, computed by convolving the sound waveform with filters designed to mimic the pseudo-logarithmic frequency resolution of cochlear responses (*McDermott and Simoncelli, 2011*). The cochleagram for each sound was composed of the compressed envelopes of these filter responses (compression is designed to mimic the effects of cochlear amplification at low sound levels). Modulation statistics were measured from filtered cochleagrams, computed by convolving each cochleagram in time and frequency with a filter designed to highlight modulations at a particular temporal rate and/or spectral scale (*Chi et al., 2005*). The temporal and spectral modulation filters were only modulated in time or frequency, respectively. There were nine temporal filters (best rates: 0.5, 1, 2, 4, 8, 16, 32, 64, and 128 Hz) and six spectral filters (best scales: 0.25, 0.5, 1, 2, 4, and 8 cycles per octave). Spectrotemporal filters

were created by taking the outer product of all pairs of temporal and spectral filters in the 2D Fourier domain, which results in oriented, gabor-like filters.

Our synthesis algorithm matches time-averaged statistics of the cochleagrams and filtered cochleagrams via a histogram-matching procedure that implicitly matches all time-averaged statistics of the responses (separately for each frequency channel of the cochleagrams and filtered cochleagrams). This choice is motivated by the fact that both fMRI and fUS reflect time-averaged measures of neural activity because the temporal resolution of hemodynamic changes is much slower than the underlying neuronal activity. As a consequence, if the fMRI or fUS response is driven by a particular set of acoustic features, we would expect two sounds with similar time-averaged statistics for those features to yield a similar response. We can therefore think of the natural and synthetic sounds as being matched under a particular model of the fMRI or fUS response (a formal derivation of this idea is given in *Norman-Haignere et al., 2018*).

We note that the filters used to compute the cochleagram were designed to match the frequency resolution of the human cochlea, which is thought to be somewhat finer than the frequency resolution of the ferret cochlea (*Walker et al., 2019*). In general, synthesizing sounds from broader filters results in synthetics that differ slightly more from the originals. And thus if we had used cochlear filters designed to mimic the frequency tuning of the ferret cochlea, we would expect the cochlear-matched synthetic sounds to differ slightly more from the natural sounds. However, given that we already observed highly divergent responses to natural and cochlear-matched synthetic sounds in both species, it is unlikely that using broader cochlear filters would change our findings. In general, we have found that the matching procedure is not highly sensitive to the details of the filters used. For example, we have found that sounds matched on the spectrotemporal filters used here and taken from *Chi et al., 2005* are also well matched on filters with half the bandwidth, with phases that have been randomized, and with completely random filters (*Norman-Haignere et al., 2018*).

## Stimuli for experiment II

Experiment II tested a larger set of 30 ferret vocalizations (5 fight calls, 17 single-pup calls, and 8 multi-pup calls where the calls from different pups overlapped in time). The vocalizations consisted of recordings from several labs (our own, Stephen David's and Andrew King's laboratories). For comparison, we also tested 14 speech sounds and 16 music sounds, yielding 60 natural sounds in total. For each natural sound, we created a synthetic sound matched on the full spectrotemporal model. We did not synthesize sounds for the sub-models (cochlear, temporal modulation, and spectral modulation) since our goal was to test if there were divergent responses to natural and synthetic ferret vocalizations for spectrotemporally matched sounds, like those present in human non-primary auditory cortex for speech and music sounds.

## Procedure for presenting stimuli and measuring voxel responses

Sounds were played through calibrated earphones (Sennheiser IE800 earphones, HDVA 600 amplifier, 65 dB) while recording hemodynamic responses via fUS imaging. In our prior fMRI experiments in humans, we had to chop the 10 s stimuli into 2 s excerpts to present the sounds in between scan acquisitions because MRI acquisitions produce a loud sound that would otherwise interfere with hearing the stimuli. Because fUS imaging produces no audible noise, we were able to present the entire 10 s sound without interruption. The experiment was composed of a series of 20 s trials, and fUS acquisitions were synchronized to trial onset. On each trial, a single 10 s sound was played, with 7 s of silence before the sound to establish a response baseline, and 3 s of post-stimulus silence to allow the response to return to baseline. There was a randomly chosen 3–5 s gap between each trial. Sounds were presented in random order, and each sound was repeated four times.

Like fMRI, the response timecourse of each fUS voxel shows a gradual build-up of activity after a stimulus due to the slow and gradual nature of blood flow changes. The shape of this response timecourse is similar across different sounds, but the magnitude varies (*Figure 1C*) (fMRI responses show the same pattern). We therefore measured the response magnitude of each voxel by averaging the response to each sound across time (from 3 to 11 s post-stimulus onset; results were robust to the particular time window used), yielding one number per sound. Before this step, we normalized responses by the prestimulus baseline for each voxel in order to account for differences in voxel perfusion levels. Specifically, we removed the mean baseline signal for each trial and then divided by the

mean baseline signal across the whole session. Responses were measured from denoised data. We describe the denoising procedure at the end of Materials and methods because it is more involved than our other analyses.

## Procedure for presenting stimuli in humans

The human data collection procedures have been described in detail previously (*Norman-Haignere et al., 2018*). Here, we give a brief overview, noting aspects of the design that are relevant to understanding the analyses.

Stimuli were presented using two slightly different paradigms. In paradigm I, we presented all four synthesis conditions in six subjects and three synthesis conditions in the other six subjects (the spectral modulation condition was missing). The natural sounds were presented twice per scan, but the synthetic sounds were only presented once to fit all of the stimuli into a single 2 hr scan. In paradigm II, we just tested the natural and fully matched synthetic sounds, which allowed us to repeat both sets of sounds 3–4 times per scan. Four subjects in paradigm I were scanned multiple times so that we could more reliably measure responses from their individual brain (three subjects completed five scans, one subject completed three scans). Five subjects were scanned in paradigm II (one subject was scanned in both paradigms), and all were scanned multiple times (one subject completed four scans, two subjects completed three scans, and one subject completed two scans).

fMRI scan acquisitions produce a loud noise due to rapid gradient switching. To prevent these noises from interfering with subjects' ability to hear the sounds, we used a 'sparse' scanning paradigm (*Hall et al., 1999*) that alternated between presenting sounds and acquiring scans. This was achieved by dividing each 10 s stimulus into five 2 s segments (windowed with 25 ms linear ramps). These five segments were presented sequentially with a single scan acquired after each segment. Each scan acquisition lasted 1 s in paradigm I and 1.05 s in paradigm II. There was a 200 ms buffer of silence before and after each acquisition. The total duration of each five-segment block was 17 s in paradigm I and 17.25 s in paradigm II. We averaged the responses of the second through fifth acquisitions after the onset of each stimulus block. The first acquisition was discarded to account for the hemodynamic delay.

## Mapping of tonotopic organization with pure tones

Tonotopic organization was assessed using previously described methods (*Bimbard et al., 2018*). In short, responses were measured to 2 s long pure tones from five different frequencies (602 Hz, 1430 Hz, 3400 Hz, 8087 Hz, 19,234 Hz). The tones were played in random order, with 20 trials/frequency. Data were denoised using the same method described in 'Denoising part I: removing components outside of cortex.' Tonotopic maps were created by determining the best frequency of each voxel, defined as the tone evoking the largest power Doppler response. Voxel responses were measured as the average response between 3 and 5 s after tone onset. We used a shorter window because the stimuli were shorter (2 s vs. 10 s). We then used these functional landmarks in combination with brain and vascular anatomy to establish the borders between primary and non-primary areas in all hemispheres, as well as to compare them to those obtained with natural sounds (see *Figure 3—figure supplement 2A*).

## Brain map display

Views from above were obtained by computing the average of the variable of interest in each vertical column of voxels from the upper part of the manually defined cortical mask. All of our measures were similar across depth (see *Figure 3—figure supplement 2C* for examples). We note that having a three-dimensional dataset was important to measuring responses from throughout the highly folded cortical ribbon.

## Spatial correlation analysis

We compared the precision and reliability of the fUS and fMRI data by measuring the correlation between all pairs of voxels and binning the results based on their distance (*Figure 1D* plots the mean correlation within each bin; ferret bin size was 0.5 mm; human bin size was 3 mm). The correlation was computed across two independent measurements of each voxel's response (odd vs. even repetitions). As a measure of spatial precision, we computed the distance needed for the correlation to decay by 75% :

$$f(\tau_{75}) = (1 - 0.75)f(0) \tag{1}$$

where $f(.)$ is the correlation vs. distance function and $\tau$ is the 75% decay rate, computed by solving the above equation (via linear interpolation). The human data showed an above 0 correlation at very long distances, suggesting that there is a shared response pattern present across all voxels. To prevent this baseline difference from affecting the decay rate, we first normalized the correlation by subtracting the minimum correlation across all distances before applying the above equation. We statistically compared the reliability (0 mm correlation) and 75% decay rate of the spatial correlation function across species using a Wilcoxon rank-sum test across subjects.

## NSE maps

We compared the response magnitude to natural and corresponding synthetic sounds using the NSE, the same metric used in humans. The NSE is defined as

$$NSE = \frac{\mu([xy]^2)}{\mu(x^2)+\mu(y^2)-2\mu(x)\mu(y)} \tag{2}$$

where **x** and **y** are response vectors across the sounds being compared (i.e., natural and synthetic). The squares in the above equation indicate that each element of the vector is being squared. $\mu(.)$ indicates the mean across all elements in the vector.

The NSE takes a value of 0 if the response to natural and synthetic sounds is identical and 1 if there is no correspondence between responses to natural and synthetic sounds (i.e., they are independent). For anticorrelated signals, the NSE can exceed 1 with a maximum value of 2 for signals that are zero-mean and perfectly anticorrelated. This is analogous to the correlation coefficient, which has a maximum value of 1 for identical signals, a minimum value of –1 for anticorrelated signals, and a value of 0 for independent signals.

Unlike the correlation coefficient, the NSE is sensitive to differences in the mean and scale of the responses being compared, in addition to differences in the response pattern. This property is useful because the model predicts that the responses to natural and synthetic sounds should be matched (*Norman-Haignere et al., 2018*), and thus any divergence in the response to natural vs. synthetic sounds reflects a model failure, regardless of whether that divergence is driven by the pattern, mean, or scale of the response. In ferrets, we observed NSE values near 0 throughout ferret auditory cortex, indicating that responses are approximately matched in all respects. In contrast, humans showed large NSE values in non-primary auditory cortex, which could in principle be driven by differences in the mean, scale, or response pattern. In our prior work, we showed that these high NSE values are primarily driven by stronger responses to natural vs. synthetic sounds, which manifests as a downward scaling of the response to synthetic sounds. The stronger responses to natural sounds are presumably driven by sensitivity to higher-order structure that is absent from the synthetic sounds.

We noise-corrected the NSE to prevent differences in SNR from affecting our estimates, although we note that the denoised responses were highly reliable and thus correction had relatively little effect on the measured values. We used a noise-correction procedure that we previously derived and validated in simulations (*Norman-Haignere et al., 2018*). Here, we give a brief description of the method. As is evident in the equation below (an expanded version of Equation 2), the NSE can be written as a function of three statistics: (1) the power ($u\left(x^2\right)$, $u\left(y^2\right)$); (2) mean ($u\left(x\right)$, $u\left(y\right)$); and (3) cross-product ($u\left(x \circ y\right)$) of the responses being compared.

$$NSE = \frac{u\left(x^2\right)+u\left(y^2\right)-2u\left(x \circ y\right)}{u\left(x^2\right)+u\left(y^2\right)-2u\left(x\right)u\left(y\right)} \tag{3}$$

The means and cross-products are unbiased by noise as long as the noise is zero-mean, which is a trivial assumption (e.g., if we define the noise-free signal as the average response in the limit of infinite measurements, then the noise around this average is by definition zero-mean). The response power however is biased upward by noise. We can estimate the magnitude of this upward bias by calculating the power of the residual error between two independent measurements of the response (i.e., two different repetitions of the same stimuli), which is equal to twice the noise power in expectation. By subtracting off half the residual power, we can thus noise-correct our power estimates:

$$\hat{\mu}\left(\boldsymbol{x}^2\right) = \tfrac{1}{2}\mu\left(\boldsymbol{x}_1^2\right) + \tfrac{1}{2}\mu\left(\boldsymbol{x}_2^2\right) - \tfrac{1}{2}\mu\left(\left[\boldsymbol{x}_1 - \boldsymbol{x}_2\right]^2\right) \tag{4}$$

$$\hat{\mu}\left(\boldsymbol{y}^2\right) = \tfrac{1}{2}\mu\left(\boldsymbol{y}_1^2\right) + \tfrac{1}{2}\mu\left(\boldsymbol{y}_2^2\right) - \tfrac{1}{2}\mu\left(\left[\boldsymbol{y}_1 - \boldsymbol{y}_2\right]^2\right) \tag{5}$$

where, for example, $\boldsymbol{x}_1$ and $\boldsymbol{x}_2$ are two independent measurements of the response to natural sounds and $\boldsymbol{y}_1$ and $\boldsymbol{y}_2$ are two independent measurements of the response to synthetic sounds.

We only analyzed voxels that had a test–retest NSE less than 0.4, which we previously found in simulations was sufficient to achieve reliable noise-corrected measures (*Norman-Haignere et al., 2018*). Most voxels in auditory cortex passed this threshold since the denoised voxel responses were highly reliable.

## Annular ROI analyses

We used the same annular ROI analyses from our prior paper to quantify the change in NSE values (or lack thereof) across the cortex. We binned voxels based on their distance to the center of primary auditory cortex, defined tonotopically. We used smaller bin sizes in ferrets (0.5 mm) than humans (5 mm) due to their smaller brains (results were not sensitive to the choice of bin size). *Figure 2F* plots the median NSE value in each bin, plotted separately for each human and ferret subject. To statistically compare different models (e.g., cochlear vs. spectrotemporal), for each animal, we computed the median NSE value across all voxels separately for each model, bootstrapped the resulting statistics by resampling across the sound set (1000 times), and counted the fraction of samples that overlapped between models (multiplying by 2 to arrive at a two-sided p-value). To compare species, we measured the slope of the NSE vs. distance curve for the fully matched synthetic sounds separately for each human and ferret subject. We then compared each ferret slope to the distribution of human slopes using a sign test to evaluate if that individual ferret differed significantly from the human population.

## Human analyses

The details of the human analyses very were similar to those in our prior paper (*Norman-Haignere et al., 2018*). To make the human and ferret analyses more similar, we used component-denoised fMRI data. Results were similar without denoising (*Figure 1—figure supplement 2*). Denoising was accomplished by projecting the response of each voxel of each subject onto the six reliable components inferred in our prior studies (see *Figure 3—figure supplement 3*; *Norman-Haignere et al., 2015*; *Norman-Haignere et al., 2018*).

Whole-brain NSE maps are based on data for paradigm I and were computed by simply averaging responses across voxels in standardized coordinates (FsAverage template brain distributed by Free-surfer) and applying our NSE measures to these group averaged responses. For individual subject analyses, we used all of the available data for a given condition and the number of subjects is indicated in all relevant plots. Unlike in our prior study, we were able to get reliable NSE estimates from individual subjects with just a single scan of data because of our denoising procedure. Note that some subjects were not included in the annular ROI analyses because we did not have tonotopy data for them and thus could not functionally identify the center of PAC. When using data for paradigm I, we used just the natural sounds to estimate the noise power and correct our NSE measures since only those were presented multiple times in each scan (*Norman-Haignere et al., 2018*) (note that we have no reason to expect fMRI noise to differ across stimuli).

## Component analyses

To investigate the organization of fUS responses to the sound set, we applied the same voxel decomposition used in our prior work in humans to identify a small number of component response patterns that explained a large fraction of the response variance. Like all factorization methods, each voxel is modeled as the weighted sum of a set of canonical response patterns that are shared across voxels. The decomposition algorithm is similar to standard algorithms for independent component analysis (ICA) in that it identifies components that have a non-Gaussian distribution of weights across voxels by minimizing the entropy of the weights (the Gaussian distribution has the highest entropy of any distribution with fixed variance). This optimization criterion is motivated by the fact that independent variables become more Gaussian when they are linearly mixed, and non-Gaussianity thus provides a statistical signature that can be used to unmix the latent variables. Our algorithm differs from standard

algorithms for ICA in that it estimates entropy using a histogram, which is effective if there are many voxels, as is the case with fMRI and fUS (40,882 fUS voxels for experiment I, 38,366 fUS voxels for experiment II).

We applied our analyses to the denoised response timecourse of each voxel across all sounds (each column of the data matrix contained the concatenated response timecourse of one voxel across all sounds). Our main analysis was performed on voxels concatenated across both animals tested. The results however were similar when the analysis was performed on data from each animal separately. The number of components was determined via a cross-validation procedure described in the section on denoising.

We examined the inferred components by plotting and comparing their response profiles to the natural and synthetic sounds, as well as plotting their anatomical weights in the brain. We also correlated the response profiles across all sounds with measures of cochlear and spectrotemporal modulation energy. Cochlear energy was computed by averaging the cochleagram for each sound across time. Spectrotemporal modulation energy was calculated by measuring the strength of modulations in the filtered cochleagrams (which highlight modulations at a particular temporal rate and/or spectral scale). Modulation strength was computed as the standard deviation across time of each frequency channel of the filtered cochleagram. The channel-specific energies were then averaged across frequency, yielding one number per sound and spectrotemporal modulation rate.

We used a permutation test across the sound set to assess the significance of correlations with frequency and modulation features. Specifically, we measured the maximum correlation across all frequencies and all modulation rates tested, and we compared these values with those from a null distribution computed by permuting the correspondence across sounds between the features and the component responses (1000 permutations). We counted the fraction of samples that overlapped the null distribution and multiplied by 2 to get a two-sided p-value. For every component, we found that correlations with frequency and modulation features were significant (p<0.01).

We separately analyzed responses from experiments I (*Figure 3*) and II (*Figure 4—figure supplement 2*) because there was no simple way to combine the data across experiments since the stimuli were distinct and there was no obvious correspondence across voxels since the data were collected from different slices on different days.

## Predicting human components from ferret responses

To quantify which component response patterns were shared across species, we tried to linearly predict components across species (*Figure 3—figure supplement 4*, *Figure 3—figure supplement 5*). Each component was defined by its average response to the 36 natural and corresponding synthetic sounds, matched on the full spectrotemporal model. We attempted to predict each human component from all of the ferret components and vice versa, using cross-validated ridge regression (9 folds). The ridge parameter was chosen using nested cross-validation within the training set (also 9 folds; testing a very wide range from $2^{-100}$ to $2^{100}$). Each fold contained pairs of corresponding natural and synthetic sounds so that there would be no dependencies between the train and test sounds (i.e., the natural and synthetic version of a sound could not straddle the train and test set).

For each component, we separately measured how well we could predict the response to synthetic sounds (*Figure 3—figure supplement 4B*, *Figure 3—figure supplement 5A*) – which isolates tuning for frequency and modulation statistics present in natural sounds – as well as how well we could predict the difference between responses to natural vs. synthetic sounds (*Figure 3—figure supplement 4C*, *Figure 3—figure supplement 5B*) – which isolates sensitivity for features in natural sounds that are not explained by frequency and modulation statistics. We quantified prediction accuracy using the NSE and used $\left(1 - NSE\right)^2$ as a measure of explained variance. This choice is motivated by the fact that $\left(1 - NSE\right)$ is equivalent to the Pearson correlation for signals with equal mean and variance and $\left(1 - NSE\right)^2$ is therefore analogous to the squared Pearson correlation, which is a standard measure of explained variance. We multiplied these explained variance estimates by the total response variance of each component for either synthetic sounds or for the difference between natural and synthetic sounds (*Figure 3—figure supplement 4D,E* and *Figure 3—figure supplement 5C,D* show the total variance alongside the fraction of that total variance explained by the cross-species prediction).

When possible, we noise-corrected both the NSE and the total variance to provide the best possible estimate of their true values. Results were similar without correction. We did not noise-correct the NSE

when the component responses being predicted were themselves unreliable (test–retest NSE >0.4) since that makes the correction unreliable (**Norman-Haignere et al., 2018**); this occurred, for example, when attempting to predict the natural vs. synthetic differences in ferrets for which there was virtually no reliable variance (see **Figure 3—figure supplement 5D**).

We noise-corrected the total variance using the equation below:

$$\frac{var(r_1+r_2)-var(r_1-r_2)}{4} \tag{6}$$

where $r_1$ and $r_2$ are two independent response measurements. Below, we give a brief derivation of this equation, where $r_1$ and $r_2$ are expressed as the sum of a shared signal ($s$) that is repeated across measurements plus independent noise ($n_1$ and $n_2$) which is not. This derivation utilizes the fact that the variance of independent signals that are summed or subtracted is equal to the sum of their respective variances.

$$
\begin{aligned}
\frac{var(r_1+r_2)-var(r_1-r_2)}{4} &= \frac{var([s+n_1]+[s+n_2])-var([s+n_1]-[s+n_2])}{4} \\
&= \frac{var(2s+n_1+n_2)-var(n_1-n_2)}{4} \\
&= \frac{4var(s)}{4} \\
&= var(s) \tag{7}
\end{aligned}
$$

The two independent measurements used for noise correction were derived from different human or ferret subjects. The measurements were computed by attempting to predict group components from each subject using the same cross-validated regression procedure described above. The two measurements in ferrets came from the two animals tested (A and T). And the two measurements in humans came from averaging the predictions across two non-overlapping sets of subjects (four in each group; groups chosen to have similar SNR).

For this analysis, the components were normalized so that the RMS magnitude of their weights was equal. As a consequence, components that explained more response variance also had larger response magnitudes. We also adjusted the total variance across all components to equal 1.

We computed error bars by bootstrapping across sounds (1000 samples). Specifically, we sampled sounds with replacement and then re-computed the NSE and total variance using the sampled sounds. Note that we did not allow squaring to make negative values positive (i.e., in $(1-NSE)^2$) since that would bias the distribution.

## Comparing the similarity of natural and synthetic sounds from different categories

We computed maps showing the average difference between natural and synthetic sounds from different categories (**Figure 4C**). So that the scale of the differences could be compared across species, we divided the measured differences by the standard deviation of each voxel's response across all sounds. We also separately measured the NSE for individual sounds (**Figure 4—figure supplement 1D**) and sound categories (**Figure 4C and D**). For this analysis, the numerator of the NSE (Equation 2) was computed in the normal way by measuring the error between natural and synthetic sounds for the particular sounds/categories of interest. The denominator/normalization term was computed using all sounds to ensure that the normalization was the same for all sounds/categories and thus that we were not inadvertently normalizing away meaningful differences. To statistically compare the categories, we applied a Wilcoxon rank-sum test to the distribution of NSE values across sounds from the categories being compared.

## Video recording

We measured the motion of the animal using a video recording of the animal's face (**Figure 4—figure supplement 1A and B**). Specifically, we measured the absolute value of the frame-to-frame deviations in the video and summed these differences across all pixels within an ROI centered on the ferret's face. We computed evoked movement in a similar way as for fUS signals. Specifically, we removed the mean movement during the baseline for each trial and then divided by the mean baseline movement across the whole session. We computed the average motion evoked by each sound by averaging across recording sessions, separately for each animal. Before averaging, to account for different

camera angles across recording sessions, we divided the movement by the standard deviation across sounds in each session. We statistically compared motion between different sound categories using a Wilcoxon rank-sum test across the sounds from each category.

## Denoising part I: Removing components outside of cortex

Ultrasound responses in awake animals are noisy, which has limited its usage to mapping simple stimulus dimensions (e.g., frequency) where a single stimulus can be repeated many times (*Bimbard et al., 2018*). To overcome this issue, we developed a denoising procedure that substantially increased the reliability of the voxel responses (*Figure 1—figure supplement 1*). The procedure had two parts. The first part, described in this section, removed prominent signals outside of cortex, which are likely to reflect movement or other sources of noise. The second part enhanced reliable signals. Code implementing the denoising procedures is publicly available (https://github.com/agneslandemard/naturalsounds_analysis, copy archived at swh:1:rev:89466e7b5492553d3af314b7d4fff6d059445588; *Landemard, 2021*).

We separated voxels into those inside and outside of cortex since responses outside of the cortex by definition do not contain stimulus-driven cortical responses, but do contain sources of noise like motion. We then used canonical correlation analysis (CCA) to find a set of response timecourses that were robustly present both inside and outside of cortex since such timecourses are both likely to reflect noise and likely to distort the responses of interest (*de Cheveigné et al., 2019*). We projected out the top 20 canonical components (CCs) from the dataset, which we found scrubbed the data of motion-related signals (*Figure 4—figure supplement 3*; motion described below).

This analysis was complicated by one key fact: the animals reliably moved more during the presentation of some sounds (*Figure 4—figure supplement 1B*). Thus, noise-induced activity outside of cortex is likely to be correlated with sound-driven neural responses inside of cortex, and removing CCs will thus remove both noise and genuine sound-driven activity. To overcome this issue, we took advantage of the fact that sound-driven responses will by definition be reliable across repeated presentations of the same sound, while motion-induced activity will vary from trial to trial for the same sound. We thus found CCs where the residual activity after removing trial-averaged responses was shared between responses inside and outside of cortex, and we then removed the contribution of these components from the data. We give a detailed description and motivation of this procedure in Appendix 1 and show the results of a simple simulation demonstrating its efficacy.

To assess the effect of this procedure on our fUS data, we measured how well it removed signals that were correlated with motion (*Figure 4—figure supplement 3A*). Motion was measured using a video recording of the animal's face. We measured the motion energy in the video as the average absolute deviation across adjacent frames, summed across all pixels. We correlated this motion timecourse with the timecourse of every voxel. *Figure 4—figure supplement 3A* plots the mean absolute correlation value across voxels as a function of the number of CCs removed (motion can induce both increased and decreased fUS signal, and thus it was necessary to take the absolute value of the correlation before averaging). We found that removing the top 20 CCs substantially reduced motion correlations.

We also found that removing the top 20 CCs removed spatial striping in the voxel responses, which is a stereotyped feature of motion due to the interaction between motion and blood vessels. To illustrate this effect, *Figure 4—figure supplement 3B* shows the average difference between responses to natural vs. synthetic sounds in experiment II (vocalization experiment). Before denoising, this difference map shows a clear striping pattern likely due to the fact that the animals moved more during the presentation of the natural vs. synthetic sounds. The denoising procedure largely eliminated this striping pattern.

## Denoising part II: Enhancing signal using DSS

After removing components likely to be driven by noise, we applied a second procedure designed to enhance reliable components in the data. Our procedure is a variant of a method that is often referred to as 'denoising source separation' (DSS) or 'joint decorrelation' (*de Cheveigné and Parra, 2014*). In contrast with principal component analysis (PCA), which finds components that have high variance, DSS emphasizes components that have high variance after applying a 'biasing' operation that is designed to enhance some aspect of the data. The procedure begins by whitening the data

such that all response dimensions have equal variance, the biasing operation is applied, and PCA is then used to extract the components with the highest variance after biasing. In our case, we biased the data to enhance response components that were reliable across stimulus repetitions and slices. This procedure was done for each animal independently. We note that unlike fMRI, data from different slices come from different sessions. As a consequence, the noise from different slices will be independent. Thus, any response components that are consistent across slices are likely to reflect true, stimulus-driven responses.

The input to our analysis was a set of matrices. Each matrix contained data from a single stimulus repetition and slice. Only voxels from inside of cortex were analyzed. Each column of each matrix contained the response timecourse of one voxel to all of the sounds (concatenated), denoised using the procedure described in part I. The response of each voxel was converted to units of percent signal change (the same units used for fMRI analyses) by subtracting and dividing by the pre-stimulus period (also known as percent cerebral blood volume [%CBV] in the fUS literature).

Our analysis involved five steps:

1. We whitened each matrix individually.

2. We averaged the whitened response timecourses across repetitions, thus enhancing responses that are reliable across repetitions.

3. We concatenated the repetition-averaged matrices for all slices across the voxel dimension, thus boosting signal that is shared across slices.

4. We extracted the top N principal components (PCs) with the highest variance from the concatenated data matrix. The number of components was selected using cross-validation (described below). Because the matrices for each repetition and slice have been whitened, the PCs extracted in this step will *not* reflect the components with highest variance, but will instead reflect the components that are the most reliable across repetitions and slices. We thus refer to these components as 'reliable components' ($R$).

5. We then projected the data onto the top N reliable components ($R$):

$$D_{denoised} = RR^+D \tag{8}$$

where $D$ is the denoised response matrix from part I and + indicates the matrix pseudoinverse.

We used cross-validation to test the efficacy of this denoising procedure and select the number of components (***Figure 1—figure supplement 1***). The same number of components was selected across animals. This analysis involved the following steps:

1. We divided the sound set into training (75%) and test (25%) sounds. Each set contained corresponding natural and synthetic sounds so that there would be no overlap between train and test sets. We attempted to balance the train and test sets across categories such that each split had the same number of sounds from each category.

2. Using responses to just the train sounds ($D_{train}$), we computed reliable components ($R_{train}$) using the procedure just described (steps 1–4 in the above section).

3. We calculated voxel weights for these components:

$$W = R_{train}^+ D_{train} \tag{9}$$

4. We used this weight matrix, which was derived entirely from train data, to denoise responses to the test sounds:

$$D_{test-denoised} = R_{test}W \tag{10}$$

$$R_{test} = D_{test}W^+ \tag{11}$$

To evaluate whether the denoising procedure improved predictions, we measured responses to the test sound set using two independent splits of data (odd or even repetitions). We then correlated the responses across the two splits either before or after denoising.

***Figure 1—figure supplement 1A*** plots the split-half correlation of each voxel before vs. after denoising for every voxel in the cortex (using an eight-component model). For this analysis, we either denoised one split of data (blue dots) or both splits of data (green dots). Denoising one split provides

a fairer test of whether the denoising procedure enhances SNR, while denoising both splits demonstrates the overall boost in reliability. We also plot the upper bound on the split-half correlation when denoising one split of data (black line), which is given by the square root of the split-half reliability of the original data. We found that our denoising procedure substantially increased reliability with the denoised correlations remaining close to the upper bound. When denoising both splits, the split-half correlations were near 1, indicating a highly reliable response.

*Figure 1—figure supplement 1B* plots a map in one animal of the split-half correlations when denoising one split of data along with a map of the upper bound. As is evident, the denoised correlations remain close to the upper bound throughout primary and non-primary auditory cortex.

*Figure 1—figure supplement 1C* shows the median split-half correlation across voxels as a function of the number of components. Performance was best using approximately eight components in both experiments.

## Acknowledgements

We thank Sophie Bagur for careful reading of the manuscript and precious comments.

## Additional information

### Funding

| Funder | Grant reference number | Author |
|---|---|---|
| Agence Nationale de la Recherche | ANR-17-EURE-0017 ANR-10-IDEX-0001-02 | Agnès Landemard<br>Célian Bimbard<br>Shihab Shamma<br>Yves Boubenec |
| H2020 European Research Council | 787836-NEUME | Shihab Shamma |
| National Institutes of Health | K99/R00 | Sam Norman-Haignere |
| Howard Hughes Medical Institute | | Sam Norman-Haignere |
| Life Sciences Research Foundation | Postdoctoral Fellowship | Sam Norman-Haignere |
| National Institutes of Health | NIDCD DC005779 | Shihab Shamma |
| Agence Nationale de la Recherche | ANR-JCJC-DynaMiC | Yves Boubenec |
| EMBO | ALTF 740-2019 | Célian Bimbard |

The funders had no role in study design, data collection and interpretation, or the decision to submit the work for publication.

### Author contributions

Agnès Landemard, Célian Bimbard, Conceptualization, Formal analysis, Investigation, Methodology, Visualization, Writing – original draft, Writing – review and editing; Charlie Demené, Resources, Software; Shihab Shamma, Conceptualization, Funding acquisition, Supervision, Writing – review and editing; Sam Norman-Haignere, Conceptualization, Formal analysis, Methodology, Supervision, Writing – original draft, Writing – review and editing; Yves Boubenec, Conceptualization, Funding acquisition, Methodology, Project administration, Supervision, Writing – original draft, Writing – review and editing

### Author ORCIDs

Agnès Landemard ⓘ http://orcid.org/0000-0001-6081-1014
Célian Bimbard ⓘ http://orcid.org/0000-0002-6380-5856
Sam Norman-Haignere ⓘ http://orcid.org/0000-0001-9342-6868

Yves Boubenec ⓘD http://orcid.org/0000-0002-0106-6947

### Ethics

Experiments were approved by the French Ministry of Agriculture (protocol authorization: 21022) and strictly comply with the European directives on the protection of animals used for scientific purposes (2010/63/EU).

### Decision letter and Author response

Decision letter https://doi.org/10.7554/eLife.65566.sa1
Author response https://doi.org/10.7554/eLife.65566.sa2

---

## Additional files

### Supplementary files

• Supplementary file 1. List of sounds used in both experiments. Names of sounds used in experiments I and II, grouped by category at both fine and coarse scales.

• Transparent reporting form

### Data availability

Our data is publicly available on Zenodo at the following link: https://doi.org/10.5281/zenodo.5493682 We provide ferret fUS data, before and after denoising, as well as additional files necessary to run our analyses. Source code for our denoising procedure and production of main figures is available on https://github.com/agneslandemard/naturalsounds_analysis.

The following dataset was generated:

| Author(s) | Year | Dataset title | Dataset URL | Database and Identifier |
|---|---|---|---|---|
| Landemard A, Bimbard C, Demené C, Shamma S, Norman-Haigneré S, Boubenec Y | 2021 | fUS imaging of ferret auditory cortex during passive listening of natural sounds | https://doi.org/10.5281/zenodo.4349594 | Zenodo, 10.5281/zenodo.4349594 |

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

# Appendix 1

## Recentered CCA

### Derivation

The goal of the denoising procedure described in part I was to remove artifactual components that were present both inside and outside of cortex since such components are both likely to be artifactual and likely to distort the responses of interest. The key complication was that motion-induced artifacts are likely to be correlated with true sound-driven neural activity because the animals reliably moved more during the presentation of some sounds. To deal with this issue, we used the fact that motion will vary from trial-to-trial for repeated presentations of the same sound, while sound-driven responses by definition will not. Here, we give a more formal derivation of our procedure. We refer to our method as 'recentered canonical correlation analysis' (rCCA) for reasons that will become clear below.

We represent the data for each voxel as an unrolled vector ($d_v$) that contains its response timecourse across all sounds and repetitions. We assume that these voxel responses are contaminated by a set of $K$ artifactual component timecourses $\{a_k\}$. We thus model each voxel as a weighted sum of these artifactual components plus a sound-driven response timecourse ($s_v$):

$$d_v = \sum_{k=1}^{K} a_k w_{k,v} + s_v \tag{12}$$

Actual voxel responses are also corrupted by voxel-specific noise, which would add an additional error term to the above equation. In practice, the error term has no effect on our derivation so we omit it for simplicity (we verified our analysis was robust to voxel-specific noise using simulations, which are described below).

To denoise our data, we need to estimate the artifactual timecourses $\{a_k\}$ and their weights ($w_{k,v}$) so that we can subtract them out. If the artifactual components $\{a_k\}$ were uncorrelated with the sound-driven responses ($s_v$), we could estimate them by performing CCA on voxel responses from inside and outside of cortex since only the artifacts would be correlated. However, we expect sound-driven responses to be correlated with motion artifacts, and the components inferred by CCA will thus reflect a mixture of sound-driven and artifactual activity.

To overcome this problem, we first subtract-out the average response of each voxel across repeated presentations of the same sound. This 'recentering' operation removes sound-driven activity, which by definition is the same across repeated presentations of the same sound:

$$\dot{d}_v = \sum_{k=1}^{K} \dot{a}_k w_{k,v} \tag{13}$$

where the dot above a variable indicates its response after recentering (not its time derivative). Because sound-driven responses have been eliminated, applying CCA to the recentered voxel responses should yield an estimate of the recentered artifacts ($\dot{a}_k$) and their weights ($w_{k,v}$) (note that CCA actually yields a set of components that span a similar subspace as the artifactual components, which is equivalent from the perspective of denoising). To simplify notation in the equations below, we assume this estimate is exact (i.e., CCA exactly returns $\dot{a}_k$ and $w_{k,v}$).

Since the weights ($w_{k,j}$) are the same for original ($d_v$) and recentered ($\dot{d}_v$) data, we are halfway done. All that is left is to estimate the original artifact components before recentering ($a_k$), which can be done using the original data before recentering ($d_v$). To see this, first note that canonical components (CCs) are by construction a linear projection of the data used to compute them, and thus, we can write

$$\dot{a}_k = \sum_{v=1}^{V} \dot{d}_v \beta_{k,v} \tag{14}$$

We can use the reconstruction weights ($\beta_{k,v}$) in the above equation to get an estimate of the original artifactual components by applying them to the original data before recentering:

$$a_k \approx \sum_{v=1}^{V} d_v \beta_{k,v} \tag{15}$$

To see this, we expand the above equation:

$$\sum_{v=1}^{V} d_v \beta_{k,j} = \sum_{v=1}^{V} \left( \sum_{k'=1}^{N} a_{k'} w_{k',v} + s_v \right) \beta_{k,v} \tag{16}$$

$$= \sum_{k'=1}^{N} a_{k'} \sum_{v=1}^{V} w_{k',v} \beta_{k,v} + \sum_{v=1}^{V} s_v \beta_{k,v} \tag{17}$$

The first term in the above equation exactly equals $a_k$ because $w_{k',v}$ and $\beta_{k,v}$ are by construction pseudoinverses of each other (i.e., $\sum_{v=1}^{V} w_{k',v} \beta_{k,v}$ is 1 when $k' = k$ and 0 otherwise). The second term can be made small by estimating and applying reconstruction weights using only data from outside of cortex, where sound-driven responses are weak.

We thus have a procedure for estimating both the original artifactual responses ($a_k$) and their weights ($w_{k,j}$), and can denoise our data by simply subtracting them out:

$$d_v - \sum_{k=1}^{K} a_k w_{k,v} \tag{18}$$

## Procedure

We now give the specific steps used to implement the above idea using matrix notation. The inputs to the analysis were two matrices ($D_{in}$, $D_{out}$), each of which contained voxel responses from inside and outside of cortex. Each column of each matrix contained the response timecourse of a single voxel, concatenated across all sounds and repetitions (i.e., $d_v$ in the above derivation). We also computed recentered data matrices ($\dot{D}_{in}$, $\dot{D}_{out}$) by subtracting out trial-averaged activity (i.e., $\dot{d}_v$).

CCA can be performed by whitening each input matrix individually, concatenating the whitened data matrices, and then computing the PCs of the concatenated matrices (*de Cheveigné et al., 2019*). Our procedure is an elaborated version of this basic design:

1. The recentered data matrices were reduced in dimensionality and whitened. We implemented this step using the singular value decomposition (SVD), which factors the data matrix as the product of two orthonormal matrices ($U$ and $V$), scaled by a diagonal matrix of singular values ($S$):

$$\dot{D}_{in} = \dot{U}_{in} \dot{S}_{in} \dot{V}_{in}^{T} \tag{19}$$

$$\dot{D}_{out} = \dot{U}_{out} \dot{S}_{out} \dot{V}_{out}^{T} \tag{20}$$

The reduced and whitened data were given by selecting the top 250 components and removing the diagonal S matrix:

$$\dot{D}_{in-white} = \dot{U}_{in}[:, 1:250] \dot{V}_{in}[:, 1:250]^{T} \tag{21}$$

$$\dot{D}_{out-white} = \dot{U}_{out}[:, 1:250] \dot{V}_{out}[:, 1:250]^{T} \tag{22}$$

2. We concatenated the whitened data matrices from inside and outside of cortex across the voxel dimension:

$$\dot{D}_{cat} = \left[ \dot{D}_{in-white}, \dot{D}_{out-white} \right] \tag{23}$$

3. We computed the top N PCs from the concatenated matrix using the SVD:

$$\dot{D}_{cat} = \dot{U}_{CC}\dot{S}_{CC}\dot{V}_{cc}^T \tag{24}$$

$\dot{U}_{CC}$ contains the timecourses of the CCs, ordered by variance, which provide an estimate of the artifactual components after recentering (i.e., $\dot{a}_k$). The corresponding weights (i.e., $w_{k,v}$) for voxels inside of cortex were computed by projecting the recentered data onto $\dot{U}_{CC}$ :

$$W_{\text{in}} = \dot{U}_{cc}^+\dot{D}_{in} \tag{25}$$

where + indicates the matrix pseudoinverse.

4. The original artifactual components before recentering (i.e., $a_k$) were estimated by learning a set of reconstruction weights ($B$) using recentered data from outside of cortex, and then applying these weights to the original data before recentering:

$$B = \dot{D}_{out}^+\dot{U}_{cc} \tag{26}$$

$$U_{cc} = D_{out}B \tag{27}$$

$U_{cc}$ is an estimate of the artifactual components before recentering (i.e., $a_k$).

5. Finally, we subtracted out the contribution of the artifactual components to each voxel inside of cortex, estimated by simply multiplying the component responses and weights:

$$D_{denoised} = D_{in} - U_{cc}W_{in} \tag{28}$$

## Simulation

We created a simulation to test our method. We simulated 1000 voxel responses, both inside and outside of cortex, using Equation 12. For voxels outside of cortex, we set the sound-driven responses to 0. We also added voxel-specific noise to make the denoising task more realistic/difficult (sampled from a Gaussian). Results were very similar across a variety of noise levels.

To induce correlations between the artifactual ($a_k$) and sound-driven responses ($s_v$), we forced them to share a subspace. Specifically, we computed the sound-driven responses as a weighted sum of a set of 10 component timecourses (results did not depend on this parameter), thus forcing the responses to be low-dimensional, as we found to be the case:

$$s_v = \sum_{j=1}^{10} u_j m_{j,v} \tag{29}$$

The artifactual timecourses were then computed as a weighted sum of these same 10 components timecourses plus a timecourse that was unique to each artifactual component:

$$a_k = p\sum_{j=1}^{10} u_j n_{j,k} + (1-p)\,b_k \tag{30}$$

where $p$ controls the strength of the dependence between the sound-driven and artifactual components with a value of 1 indicating complete dependence and 0 indicating no dependence. All of responses and weights ($u_j$ , $b_k$ , $m_{j,v}$ , $n_{j,k}$) were sampled from a unit-variance Gaussian. Sound-driven responses were constrained to be the same across repetitions by sampling the latent timecourses $u_j$ once, and then simply repeating the sampled values across repetitions. In contrast, a unique $b_k$ was sampled for every repetition to account for the fact that the artifacts like motion will vary from trial-to-trial. We sampled 20 artifactual timecourses using Equation 30.

We applied both standard CCA and our modified rCCA method to the simulated data. We measured the median NSE between the true and estimated sound-driven responses ($s_v$), computed using the two methods as a function of the strength of the dependence ($p$) between sound-driven and artifactual timecourses (*Appendix 1—figure 1A*). For comparison, we also plot the NSE for raw voxels (i.e., before any denoising) as well as the minimum possible NSE (noise floor) given the voxel-specific noise (which cannot possibly be removed using CCA or rCCA). When the

dependence factor ($p$) is low, both CCA and rCCA yield similarly good results, as expected. As the dependence increases, CCA performs substantially worse, while rCCA continues to perform well up until the point when the dependence becomes so strong that sound-driven and artifactual timecourses are nearly indistinguishable. Results were not highly sensitive to the number of components removed as long as the number of removed components was equal to or greater than the number of artifactual components (*Appendix 1—figure 1B*).

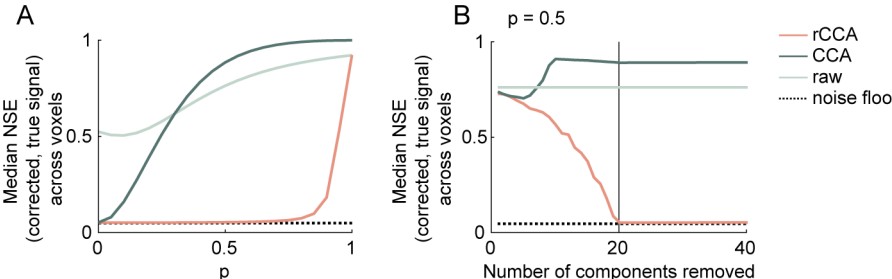

**Appendix 1—figure 1.** Simulation results. (**A**) Median normalized squared error (NSE) across simulated voxels between the true and estimated sound-driven responses ($s_v$), computed using raw/undenoised data (light green line), standard canonical correlation analysis (CCA) (dark green line), and recentered CCA (red line). Results are shown as a function of the strength of the dependence ($p$) between sound-driven and artifactual timecourses. The minimum possible NSE (noise floor) given the level of voxel-specific noise is also shown. (**B**) Same as panel (**A**), but showing results as a function of the number of components removed for a fixed value of $p$ (set to 0.5).

