## [Editor Report]

How the auditory system encodes speech sounds is not well understood, and animal models have a lot to offer in investigating such questions. This study evaluated the representations of a variety of natural and synthetic sounds in both ferrets and humans, and reported that humans differed from ferrets in the manner in which speech and music were represented, despite controlling for the spectrotemporal content of the sounds. This work makes an important contribution to our understanding of how the coding of such sounds differs across species.

---

## [Decision Letter]

**Decision letter after peer review:**

Thank you for submitting your article "Distinct higher-order representations of natural sounds in human and ferret auditory cortex" for consideration by *eLife*. Your article has been reviewed by 2 peer reviewers, and the evaluation has been overseen by a Reviewing Editor and Andrew King as the Senior Editor. The following individuals involved in review of your submission have agreed to reveal their identity: Greg Cogan (Reviewer #2); Tobias Overath (Reviewer #3).

Essential revisions:

Both reviewers found the work to be interesting and important, but the concerns about the reproducibility of the finding given the small number of animals tested weighed heavily. Given that both reviewers found merit in the work, we encourage a revised submission provided the concerns about reproducibility can be satisfactorily addressed.

*Reviewer #2 (Recommendations for the authors):*

1. Fig 2F: It would be useful here to quantify the slope as this appears to be a relevant feature of this figure.

2. Figure 2F: Are the distances for ferret vs. human chosen for a particular reason? Is it just a simple linear scaling based on brain size?

3. Is the unit size of one voxel a reasonable analysis size? If you average over all voxels in a particular region, do the results from figure 1A-D change?

4. Does the dimension reduction/component analysis (Figure 3) contain data from experiment 2 or just experiment 1? If only 1, do the results change by including the data from experiment 2?

5. While I am sure that the difference between methods of acquisition cannot fully explain your results (fUS vs. fMRI), is would be useful to comment on the relative SNR of the methods and how this would or would not influence your results.

*Reviewer #3 (Recommendations for the authors):*

1) Selectivity vs. sensitivity

The authors use the term selectivity, which implies exclusivity: e.g., response to a certain sound characteristic, but no response to any other sound characteristic. Given the actual data that the authors show, a more appropriate term to use would be sensitivity. For example, the f3 component's response profile in Figure S3 clearly shows the strongest response to speech sounds, but there is also substantial (though weaker) response to the other types of sounds. In that sense, f3 is not selective, but rather shows that it reflects maximal sensitivity to speech sounds (or, even more precisely, particular spectrotemporal characteristics of speech sounds). The authors should adjust the terminology accordingly throughout their manuscript.

2) Generalizing from 2 ferrets to all ferrets seems 'courageous' to me, especially given the replicability crisis in the human neuroimaging community. For what it's worth, the mean signal prior to denoising (Figure 1C) looks about as noisy as human fMRI data. I understand the invasive nature of fUS imaging, but I would feel much more comfortable seeing these results replicated in more animals.

3) Can the authors expand a bit on their reasoning for choosing a 3-11 s time window (line 701)? Looking at Figure 1c, it seems that this includes data from the initial rise period (which is not of interest), rather than just the (more) steady part of the response. I would have expected the authors to focus on the sustained, steady part response (e.g. 6-11 s), which presumably best reflects processing of the summary statistics of the input sound. The authors should show that their results are insensitive to (reasonable) variations in the time window.

4) NSE. I implemented the NSE formula in Matlab via

x = rand(1,40);

y = rand(1,40);

NSE = mean((x-y).^2) / (mean(x.^2) + mean(y.^2) – 2*mean(x)*mean(y))

However, the values I get for this implementation are not bounded between 0 and 1. Perhaps my implementation is wrong, or there is an error in the formula?

Also, after clarifying their NSE measure (or pointing out the mistake in the above implementation), can the authors elaborate on how NSE can distinguish between the cases (A) where a voxel has different response profiles for natural vs. model-matched sounds (e.g. x = 1:40; y = 40:-1:1;) vs. (B) where the response difference between natural and model-matched sounds is simply additive (or multiplicative) in nature (e.g. x = 1:40; y = x*2), vs. (C) when they are anticorrelated (x = [1 -1 1 -1]; y = [-1 1 -1 1])?

---

## [Author Response]

Essential revisions:Both reviewers found the work to be interesting and important, but the concerns about the reproducibility of the finding given the small number of animals tested weighed heavily. Given that both reviewers found merit in the work, we encourage a revised submission provided the concerns about reproducibility can be satisfactorily addressed.Reviewer #2 (Recommendations for the authors):1. Fig 2F: It would be useful here to quantify the slope as this appears to be a relevant feature of this figure.

Thank you for this suggestion. As noted in our general response to the editor and all reviewers, we now plot slopes for all individual human subjects and ferrets for both Experiment I (Figure 2G) and Experiment II (Figure 4F). See our note at the beginning of this response for details.

2. Figure 2F: Are the distances for ferret vs. human chosen for a particular reason? Is it just a simple linear scaling based on brain size?

The x-axis plots distances in millimeters, but in order to show both ferrets and humans on the same plot we had to rescale the axes. The corresponding unit is shown for both humans and ferrets below the axis. The 10x rescaling corresponds loosely to the difference in the radius of primary auditory cortex across species. However, our results are not sensitive to this scaling factor. Indeed, even if we use absolute distances without any rescaling, as was done to quantify the slope (Figure 2G), we still observe much more prominent changes in humans compared with ferrets. Using absolute distances substantially biases against our findings, since correcting for brain size would differentially inflate the ferret slopes relative to the human slopes. We have clarified this point in the Results (page 5):

“We used absolute distance for calculating the slopes, which is a highly conservative choice given our findings, since correcting for brain size would enhance the slopes of ferrets relative to humans. Despite this conservative choice, we found that the slope of every ferret was well below that of all 12 human subjects tested, and thus significantly different from the human group via a non-parametric sign test (p < 0.001).”

We have clarified the scaling factor in the legend of Figure 2F:

“The ferret and human data were rescaled so they could be plotted on the same figure, using a scaling factor of 10, which roughly corresponds to the difference in the radius of primary auditory cortex between ferrets and humans. The corresponding unit is plotted on the x-axis below.”

3. Is the unit size of one voxel a reasonable analysis size? If you average over all voxels in a particular region, do the results from figure 1A-D change?

In Figure 2—figure supplement 1, we plot results averaged across all voxels within standard anatomical regions-of-interest (ROIs). The results are very similar to those we find in individual voxels, with closely matched responses to natural and synthetic sounds. This result is expected since we found that individual voxels had closely matched responses and if a set of voxels have matched responses, then their average response must also be matched. We note however that the opposite is not true: if an ROI shows a matched response, there is no guarantee that the individual voxels have matched responses, since averaging across voxels could wash out heterogeneous and divergent response patterns. Thus, we believe it is appropriate to analyze individual voxels. We also note that one reason for averaging responses within an ROI is that individual voxel responses are typically quite noisy. Our denoising method however substantially boosted the reliability of our voxel responses (Figure 2 - figure supplement 1), which made it possible to analyze individual voxels.

We have clarified these points in the Results when describing our single voxel analyses (page 4):

“Our denoising procedure substantially boosted the SNR of the measurements (Figure 1 - figure supplement 1) and made it possible to analyze individual voxels, as opposed to averaging responses across a large region-of-interest (ROI), which could potentially wash out heterogeneity present at the single voxel level. […] (results were similar when averaging responses within anatomical regions of interest, see Figure 2 - figure supplement 1).”

4. Does the dimension reduction/component analysis (Figure 3) contain data from experiment 2 or just experiment 1? If only 1, do the results change by including the data from experiment 2?

Figure 3 shows results from applying component analyses to data from Experiment I. Results are very similar to Experiment II. We have added a supplemental figure (Figure 4 - figure supplement 2), which shows components from Experiment II that responded preferentially to low frequencies (top panel), high-frequencies (middle panel), and speech (bottom panel).

We note that it is not straightforward to combine data from the two experiments because the stimuli are different and the voxels are not identical because the data were recorded on different days using different slices. We have clarified this point in the Methods (page 19):

“We separately analyzed responses from Experiment I (Figure 3) and Experiment II (Figure 4 - figure supplement 2) because there was no simple way to combine the data across experiments, since the stimuli were distinct and there was no simple correspondence across voxels since the data were collected from different slices on different days.”

5. While I am sure that the difference between methods of acquisition cannot fully explain your results (fUS vs. fMRI), is would be useful to comment on the relative SNR of the methods and how this would or would not influence your results.

We have performed a new analysis to directly address this question, the results of which are shown in Figure 1D, reproduced below. We measured the correlation between pairs of fUS or fMRI voxels as a function of their distance using two independent measures of each voxel’s response (odd vs. even repetitions). As a consequence, the 0-mm datapoint provides a measure of test-retest reliability (i.e. SNR) and the fall-off with distance provides a measure of spatial precision. Results are shown separately before and after applying our component denoising method. As is evident, our denoising procedure substantially boosts the reliability of the data, which made it possible to analyze individual fUS voxels, which had low reliability before denoising. The reliability of the denoised fUS data is substantially higher than the fMRI data used in our prior study, which were not denoised, since the voxels were reliable enough to perform all of our key analyses. This finding suggests that the denoised fUS data should be more than reliable enough to detect the kinds of effects we observed previously with fMRI in humans. To make the fUS and fMRI analyses more similar, we now use component-denoised fMRI data which had similar reliability to the denoised fUS data, but our findings did not depend on this choice (see Figure 1 - figure supplement 2 if interested, which shows that results are similar for raw and denoised fMRI data).

The second noteworthy feature of this plot is that the correlation falls off more sharply for the fUS data (note the different x-axes), which we quantified by measuring the distance needed for the correlation to drop by 75% (τ75,h = 9.3 mm vs τ75,f 1.2 mm, Wilcoxon rank-sum test across subjects, p < 0.05). This plot shows fMRI data smoothed with a 5 mm FWHM kernel, which is the same kernel we used in our prior study, but the fMRI data is still substantially coarser when not smoothed (τ75,h = 6.5 mm vs τ75,f 1.2 mm, Wilcoxon rank-sum test across subjects, p < 0.05). Our human findings were very similar across different smoothing levels, indicating that the organization we detected in humans does not depend sensitively on the spatial precision of the method. These analyses suggest that our denoised fUS data is sufficiently reliable and precise to observe the kinds of functional organization we observed in humans were that organization present in ferrets.

We have added the above figure to the manuscript. The analysis is described briefly in the Results (page 4):

“We found that the denoised fUS responses were substantially more reliable and precise than the fMRI voxels from our prior study (Figure 1D) (Test-retest correlation: 0.93 vs 0.44, Wilcoxon rank-sum test across subjects, p < 0.01). To make our human and ferret analyses more similar, we used component-denoised fMRI data in this study, which had similar reliability to the denoised fUS data (Figure 1D; results were similar without denoising, see Figure 1 - figure supplement 2).”

More detail is given in the legend (above) and Methods (page 16):

“We compared the precision and reliability of the fUS and fMRI data by measuring the correlation between all pairs of voxels and binning the results based on their distance (Figure 1D plots the mean correlation within each bin; ferret bin size was 0.5 mm; human bin size was 3 mm). […] We statistically compared the reliability (0-mm correlation) and decay rate of the spatial correlation function across species using a Wilcoxon rank-sum test across subjects.”

Finally, we have included a paragraph in the Discussion that enumerates the reasons why we believe our findings are unlikely to be due to methodological differences (page 9):

“The species differences we observed are unlikely to be driven by differences in the method used to record brain responses (fUS vs. fMRI) for several reasons. […] We quantified this change by measuring the slope of the NSE-vs-distance curve and found that the slopes in ferrets were close to zero and differed substantially from every human subject tested.”

Reviewer #3 (Recommendations for the authors):1) Selectivity vs. sensitivityThe authors use the term selectivity, which implies exclusivity: e.g., response to a certain sound characteristic, but no response to any other sound characteristic. Given the actual data that the authors show, a more appropriate term to use would be sensitivity. For example, the f3 component's response profile in Figure S3 clearly shows the strongest response to speech sounds, but there is also substantial (though weaker) response to the other types of sounds. In that sense, f3 is not selective, but rather shows that it reflects maximal sensitivity to speech sounds (or, even more precisely, particular spectrotemporal characteristics of speech sounds). The authors should adjust the terminology accordingly throughout their manuscript.

We have largely removed the word “selectivity” from the manuscript and now use terms like “sensitivity” or “speech-preferring”.

2) Generalizing from 2 ferrets to all ferrets seems 'courageous' to me, especially given the replicability crisis in the human neuroimaging community. For what it's worth, the mean signal prior to denoising (Figure 1C) looks about as noisy as human fMRI data. I understand the invasive nature of fUS imaging, but I would feel much more comfortable seeing these results replicated in more animals.

See our note at the very beginning of this response which describes how we have addressed this important concern.

3) Can the authors expand a bit on their reasoning for choosing a 3-11 s time window (line 701)? Looking at Figure 1c, it seems that this includes data from the initial rise period (which is not of interest), rather than just the (more) steady part of the response. I would have expected the authors to focus on the sustained, steady part response (e.g. 6-11 s), which presumably best reflects processing of the summary statistics of the input sound. The authors should show that their results are insensitive to (reasonable) variations in the time window.

Author response image 1 shows NSE maps for two different windows. The results are virtually identical. In general, the results are highly robust to the exact window used.

**Author response image 1. sa2fig1:** 

We have clarified this point in the Methods (page 15):“We therefore measured the response magnitude of each voxel by averaging the response to each sound across time (from 3 to 11 seconds post-stimulus onset; results were robust to the window size), yielding one number per sound.”

There are already a large number of supplemental figures, but we would be happy to add this figure to supplemental if you feel it’s important.

4) NSE. I implemented the NSE formula in Matlab viax = rand(1,40);y = rand(1,40);NSE = mean((x-y).^2) / (mean(x.^2) + mean(y.^2) – 2*mean(x)*mean(y))However, the values I get for this implementation are not bounded between 0 and 1. Perhaps my implementation is wrong, or there is an error in the formula?

We greatly appreciate you for taking the time to investigate the properties of the NSE measure. Your implementation is correct and we apologize for lack of clarity. The NSE is bounded between 0 and 2, but has an expected value of 1 for independent signals with no dependency, which in most scenarios is the de facto null/upper-bound – in the same way that zero is typically the de facto null when comparing two signals even though anti-correlations are possible. Below, we include MATLAB code, which demonstrates this fact for large samples, for which the measured NSE approaches its expected value (for smaller samples, there will of course be more variation around the expected value):

N = 100000;

x = rand(1,N);

y = rand(1,N);

NSE = mean((x-y).^2) / (mean(x.^2) + mean(y.^2) – 2*mean(x)*mean(y))

The NSE can take a larger value if the signals are anticorrelated. For example, if we have two zero-mean signals that are inverses of each other, then the NSE is exactly 2:

N = 100000;

x = rand(1,N);

x = x – mean(x);

y = -x;

NSE = mean((x-y).^2) / (mean(x.^2) + mean(y.^2) – 2*mean(x)*mean(y))

This is analogous to the correlation coefficient which has a maximal value of 1 for identical signals, -1 for anticorrelated signals, and 0 for independent signals (in expectation).

We have clarified this point in the Methods (page 17):

“The NSE takes a value of 0 if the response to natural and synthetic sounds is identical and 1 if there is no correspondence between responses to natural and synthetic sounds (i.e. they are independent). For anticorrelated signals, the NSE can exceed 1 with a maximum value of 2 for signals that are zero-mean and perfectly anti-correlated. This is analogous to the correlation coefficient, which has a maximum value of 1 for identical signals, a minimum value of -1 for anticorrelated signals, and a value of 0 for independent signals.”

Also, after clarifying their NSE measure (or pointing out the mistake in the above implementation), can the authors elaborate on how NSE can distinguish between the cases (A) where a voxel has different response profiles for natural vs. model-matched sounds (e.g. x = 1:40; y = 40:-1:1;) vs. (B) where the response difference between natural and model-matched sounds is simply additive (or multiplicative) in nature (e.g. x = 1:40; y = x*2), vs. (C) when they are anticorrelated (x = [1 -1 1 -1]; y = [-1 1 -1 1])?

The NSE value is a summary measure that takes a value of 0 if the responses are identical and a higher value if the response diverges in any way, whether that be due to differences in the mean, scale, or response pattern.

As noted above, the NSE is 1 if the two responses are independent (irrespective of mean or scale):

N = 100000;

x = rand(1,N);

y = rand(1,N);

NSE = mean((x-y).^2) / (mean(x.^2) + mean(y.^2) – 2*mean(x)*mean(y))

Mean and scale differences both cause a rise in NSEs with values approaching 1 as the means and scales diverge even if the response pattern is identical:

N = 100000;

x = rand(1,N);

y = x + 1000;

NSE = mean((x-y).^2) / (mean(x.^2) + mean(y.^2) – 2*mean(x)*mean(y))

y = x*1000;

NSE = mean((x-y).^2) / (mean(x.^2) + mean(y.^2) – 2*mean(x)*mean(y))

The primary difference between the NSE and the correlation coefficient is that the correlation coefficient is insensitive to mean and scale. This property is problematic because the model predicts that the responses should be the same if the model is accurate and thus any divergence, whether it be due to mean, scale, or pattern, reflects a model failure. In ferrets, the NSE values are near 0 for fully matched synthetic sounds, which guarantees that the mean, scale, and pattern are all similar. For humans, the NSE values are large in non-primary regions which indicates a divergent response, but does not say anything about whether it is the mean, scale or pattern that differs. In our prior paper, we showed that these high NSE values in humans are primarily driven by stronger responses to natural vs. synthetic sounds, which manifests as a downward scaling of the responses to synthetic sounds. We have clarified all of these points in the Methods when describing the NSE (page 17):

“Unlike the correlation coefficient, the NSE is sensitive to differences in the mean and scale of the responses being compared, in addition to differences in the response pattern. This property is useful because the model predicts that the responses to natural and synthetic sounds should be matched (Norman-Haignere and McDermott, 2018), and thus any divergence in the response to natural vs. synthetic sounds reflects a model failure, regardless of whether that divergence is driven by the pattern, mean, or scale of the response. In ferrets, we observed NSE values near 0 throughout ferret auditory cortex, indicating that responses are approximately matched in all respects. In contrast, humans showed large NSE values in non-primary auditory cortex, which could in principle be driven by differences in the mean, scale, or response pattern. In our prior work, we showed that these high NSE values are primarily driven by stronger responses to natural vs. synthetic sounds, which manifests as a downward scaling of the response to synthetic sounds. The stronger responses to natural sounds are presumably driven by sensitivity to higher-order structure that is absent from the synthetic sounds.”